# Amortized Active Generation of Pareto Sets

**Daniel M. Steinberg**[1][*]   **Asiri Wijesinghe**[1]   **Rafael Oliveira**[1]
**Piotr Koniusz**[1,2,3]   **Cheng Soon Ong**[1,3]   **Edwin V. Bonilla**[1]
[1]CSIRO's Data61   [2]University of New South Wales   [3]Australian National University

## Abstract

We introduce active generation of Pareto sets (A-GPS), a new framework for online discrete black-box multi-objective optimization (MOO). A-GPS learns a generative model of the Pareto set that supports a-posteriori conditioning on user preferences. The method employs a class probability estimator (CPE) to predict non-dominance relations and to condition the generative model toward high-performing regions of the search space. We also show that this non-dominance CPE implicitly estimates the probability of hypervolume improvement (PHVI). To incorporate subjective trade-offs, A-GPS introduces *preference direction vectors* that encode user-specified preferences in objective space. At each iteration, the model is updated using both Pareto membership and alignment with these preference directions, producing an amortized generative model capable of sampling across the Pareto front without retraining. The result is a simple yet powerful approach that achieves high-quality Pareto set approximations, avoids explicit hypervolume computation, and flexibly captures user preferences. Empirical results on synthetic benchmarks and protein design tasks demonstrate strong sample efficiency and effective preference incorporation.

## 1   Introduction

In many scientific and engineering domains, practitioners face the challenge of optimizing complex, high-dimensional, discrete objects under expensive black-box evaluation processes. Examples include designing protein sequences for enhanced stability and activity, synthesizing small molecules with tailored pharmacokinetics, and engineering DNA constructs for precise gene regulation. In these settings, each candidate design must be evaluated via computationally intensive simulations or laboratory assays, making efficient search strategies essential. Furthermore, these applications frequently involve multiple, often conflicting objectives. For instance, in protein engineering one may wish to maximize thermal stability, catalytic turnover rate, and expression yield, yet improvements in one property can degrade another. The set of non-dominated trade-off designs (where no objective can be improved without sacrificing performance in at least one other objective) is known as the Pareto set. Accurately approximating this Pareto set is critical for enabling informed decision-making in downstream experimental workflows.

Traditional Bayesian optimization (BO) methods for black-box multi-objective optimization (MOO) problems, often referred to as multi-objective Bayesian optimization (MOBO), rely on acquisition functions such as expected hypervolume improvement (EHVI) [42, 10, 2] or their quasi-MonteCarlo extensions [11, 2], which involve complex numerical integration and/or scale poorly with the number of objectives. Alternatively they may rely on random scalarizations [27, 31, 13] offering simplicity and scalability but rely on sufficient sampling density to capture complex Pareto front geometries.

In this work we propose a fundamentally different approach, in the spirit of multi-objective generation (MOG) [44, 43], that directly estimates a generative model of the Pareto set in an online sequential,

---

[*]corresponding author: dan.steinberg@data61.csiro.au

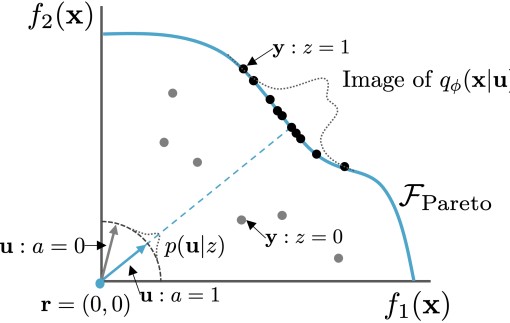

Figure 1: A visualization of a Pareto front, $\mathcal{F}_{\text{Pareto}}$, and the random variables used with A-GPS. $\mathbf{y}$ are noisy realizations of the objectives, $\mathbf{f}_\bullet$. When $z = 1$ these observations lie on the Pareto front. Preference direction vectors, $\mathbf{u}$, are unit vectors pointing to a region of the Pareto front from a reference point, $\mathbf{r}$, Equation 11. We derive aligned ($a = 1$) training preference direction vectors, $\mathbf{u}_n$, from observation pairs $(\mathbf{y}_n, \mathbf{x}_n)$, and mis-aligned preference direction vectors from permuting these pairs, $(\mathbf{y}_{\rho(n)}, \mathbf{x}_n)$, Equation 18. The aim is to learn the distribution of the Pareto set $q_\phi(\mathbf{x}|\mathbf{u}) \approx p(\mathbf{x}|\mathbf{u}, z = 1, a = 1)$.

black-box optimization setting. To this end, we build upon the recently proposed variational search distributions (VSD) framework for single-objective optimization [39]. VSD formulates optimization as active learning of a generative model of high-performing designs (active generation). Thus, VSD alternates between fitting a class probability estimator (CPE) to discriminate favorable designs and updating a conditional generative model to propose new candidates directly. Hence, instead of using a fitness-threshold probability of improvement (PI)-CPE, we propose a Pareto-set CPE, which enables the generative model to focus exclusively on non-dominated designs. This approach bypasses explicit hypervolume computations and scalarizations and leads to a scalable online algorithm for Pareto set approximation that is different from other recent MOG methods [44, 43]. Furthermore, we show that such a non-dominance CPE is implicitly estimating probability of hypervolume improvement (PHVI). Additionally, practical decision-making often requires incorporating user or stakeholder preferences over the trade-offs among objectives. We introduce a novel mechanism for embedding subjective preference conditioning into the generative model, allowing practitioners to sample candidates that align with specified trade-off directions. By using amortized variational inference (VI) with *preference direction vectors*, our method supports a-posteriori preference specification without retraining, offering flexibility in downstream design exploration.

We show that our method, active generation of Pareto sets (A-GPS), performs well against competing approaches on a suite of challenging synthetic and real multi-objective optimization benchmarks.

## 2 Preliminaries

### 2.1 Active generation

Active generation as implemented by [39] reframes online black-box optimization as sequential learning of a conditional generative model, guided by a CPE. At each round, $t \in \{1, \ldots, T\}$ we: (1) fit a CPE (using some proper loss, $\mathcal{L}_{\text{CPE}}$),

$$\pi_\theta^z(\mathbf{x}) \approx p(z = 1|\mathbf{x}), \tag{1}$$

parameterized by $\theta$ and where $z = \mathbb{1}[\mathbf{x} \in \mathcal{S}]$ indicates membership in some desired set, $\mathcal{S}$. For example, designs fitter than some incumbent, $\mathbf{x}^\star$, under some black box function, $f_\bullet : \mathcal{X} \to \mathbb{R}$; $\mathcal{S} := \{\mathbf{x} \in \mathcal{X} : f_\bullet(\mathbf{x}) > f_\bullet(\mathbf{x}^\star)\}$. Then (2) update the generative model $q_\phi(\mathbf{x})$, e.g. by minimizing the reverse Kullback-Leibler (KL) divergence to the ideal conditional, $p(\mathbf{x}|z = 1)$, or equivalently maximizing the evidence lower bound (ELBO),

$$\mathcal{L}_{\text{ELBO}}(\phi, \theta) = \mathbb{E}_{q_\phi(\mathbf{x})}[\log \pi_\theta^z(\mathbf{x})] - \mathbb{D}_{\text{KL}}[q_\phi(\mathbf{x})\|p(\mathbf{x}|\mathcal{D}_0)], \tag{2}$$

where $p(\mathbf{x}|\mathcal{D}_0)$ is a prior over the design space. Formally, using data $\mathcal{D}_N^z = \{(\mathbf{x}_n, z_n)\}_{n=1}^N$, active generation optimizes,

$$\theta_t^* \leftarrow \underset{\theta}{\arg\min} \, \mathcal{L}_{\text{CPE}}(\theta, \mathcal{D}_N^z) \qquad \phi_t^* \leftarrow \underset{\phi}{\arg\max} \, \mathcal{L}_{\text{ELBO}}(\phi, \theta_t^*), \tag{3}$$

then samples from $q_{\phi_t^*}(\mathbf{x})$ are used to propose new candidates for evaluation. New labels are acquired for these candidates, the dataset is augmented, and the process is repeated until convergence. This solution to active generation is referred to as VSD [39]. Under certain assumptions on the form of the models, this procedure has proven convergence rates to the ideal $p(\mathbf{x}|z = 1)$.

## 2.2 Optimizing over multiple objectives

In this work we are concerned with generating discrete or mixed discrete-continuous designs, for example sequences $\mathbf{x} \in \mathcal{X} = \mathcal{V}^M$, where $\mathcal{V}$ is the sequence vocabulary and $M$ is the sequence length, that have particular measurable properties $\mathbf{y} \in \mathbb{R}^L$. We assume the 'black-box' relationship $\mathbf{y} = \mathbf{f_\bullet}(\mathbf{x}) + \boldsymbol{\epsilon}$ where $\mathbf{f_\bullet}(\mathbf{x}) = [f_\bullet^1(\mathbf{x}), \ldots, f_\bullet^l(\mathbf{x}), \ldots, f_\bullet^L(\mathbf{x})]$ and $\mathbb{E}_{p(\boldsymbol{\epsilon})}[\boldsymbol{\epsilon}] = \mathbf{0}$. The black-box function $\mathbf{f_\bullet}(\cdot)$ could be a noisy empirical observation, or an expensive physics/chemistry simulation (where $\boldsymbol{\epsilon} = \mathbf{0}$), etc. In MOO we would like to find the global optimum,

$$\max_{\mathbf{x} \in \mathcal{X}} \mathbf{f_\bullet}(\mathbf{x}). \tag{4}$$

There are a number of issues that present themselves here though. Firstly, we cannot use gradient based optimization methods directly since we cannot access $\nabla_{\mathbf{x}} f_\bullet^l(\mathbf{x})$ as $f_\bullet^l$ are black-boxes and $\mathbf{x}$ is (partially) discrete. But more importantly, $\max$ is not uniquely defined for the vector valued $\mathbf{f_\bullet}$ as the individual objectives can be in conflict with one (or more formally, there is no total ordering). Instead, we are interested in finding the set of designs for which we cannot increase one objective without compromising others. This is known as the Pareto set $\mathcal{S}_{\text{Pareto}}^* \subset \mathcal{X}$,

$$\mathcal{S}_{\text{Pareto}}^* := \{\mathbf{x} : \mathbf{x}' \not\succ \mathbf{x}, \ \forall \mathbf{x}' \in \mathcal{X}\}, \tag{5}$$

where $\mathbf{x}' \succ \mathbf{x}$ refers to $\mathbf{x}'$ dominating $\mathbf{x}$, i.e., all of the objective function values for $\mathbf{x}'$ are greater than or equal to those of $\mathbf{x}$, and at least one is greater,

$$\mathbf{x}' \succ \mathbf{x} \text{ iff } f_\bullet^l(\mathbf{x}') \geq f_\bullet^l(\mathbf{x}) \ \forall l \in \{1, \ldots, L\} \text{ and } \exists l \in \{1, \ldots, L\} \text{ such that } f_\bullet^l(\mathbf{x}') > f_\bullet^l(\mathbf{x}). \tag{6}$$

The Pareto set also induces the Pareto front, $\mathcal{F}_{\text{Pareto}}^* := \{\mathbf{f_\bullet}(\mathbf{x}) : \forall \mathbf{x} \in \mathcal{S}_{\text{Pareto}}^*\}$, which is the image of the Pareto set outcomes in $\mathbb{R}^L$. For data $\mathcal{D}_N = \{(\mathbf{y}_n, \mathbf{x}_n)\}_{n=1}^N$ at round $t$ we define the current *observable* Pareto set,

$$\mathcal{S}_{\text{Pareto}}^t := \{\mathbf{x}_i : \mathbf{x}_j \not\succ \mathbf{x}_i, \ \forall i \in \{1, \ldots, N\}, \ \forall j \in \{1, \ldots, N\} \backslash i\}, \tag{7}$$

$$\text{where} \quad \mathbf{x}_j \succ \mathbf{x}_i \text{ iff } y_j^l \geq y_i^l \ \forall l \in \{1, \ldots, L\} \text{ and } \exists l \in \{1, \ldots, L\} \text{ such that } y_j^l > y_i^l.$$

We define a Pareto set membership (or non-dominance) label $z_n := \mathbb{1}[\mathbf{x}_n \in \mathcal{S}_{\text{Pareto}}^t]$, where $\mathbb{1}[\cdot] : \{\text{True}, \text{False}\} \rightarrow \{1, 0\}$, which we will use for training the CPE in active generation. This definition can be extended to the whole domain $\mathcal{X}$ as a labeling function $z(\mathbf{x}) := \mathbb{1}[\mathbf{x} \in \text{Pareto}(\mathcal{S}_{\text{Pareto}}^t \cup \{\mathbf{x}\})]$, where $\text{Pareto}(\mathcal{S})$ denotes the Pareto subset of an arbitrary set $\mathcal{S} \subset \mathcal{X}$. Note that the definition remains unchanged for observed points already in the dataset, i.e., $z(\mathbf{x}_n) = z_n$. As it turns out, this non-dominance CPE is also estimating the probability of hypervolume improvement (PHVI) for new query points as the following theorem and corollary (assuming $\boldsymbol{\epsilon} = \mathbf{0}$). We fully define hypervolume improvement (HVI) in Sec. B.1, as its definition is a little involved.

**Theorem 1** (Equivalence of Indicators). *For every $\mathbf{x} \notin \mathcal{S}_{Pareto}^t$, the HVI indicator is equivalent to a non-dominance indicator,*

$$\mathbb{1}[\text{HVI}(\mathbf{x}) > 0] = z(\mathbf{x}). \tag{8}$$

**Corollary 1** (Non-Dominance CPE estimates PHVI). *Following straightforwardly from Theorem 1,*

$$\mathbb{P}(z(\mathbf{x}) = 1|\mathbf{x}) = \mathbb{P}(\text{HVI}(\mathbf{x}) > 0|\mathbf{x}) =: \text{PHVI}(\mathbf{x}), \quad \forall \mathbf{x} \notin \mathcal{S}_{Pareto}^t, \tag{9}$$

*as the events are equivalent. Thus, a CPE trained on $z$, using a proper loss, is predicting PHVI.* □

See Sec. B.1 for the proof and assumptions under which this is true. We note that, for existing $\mathbf{x} \in \mathcal{S}_{\text{Pareto}}^t$, we have a discrepancy, as $z(\mathbf{x}) = 1$, whereas $\text{HVI}(\mathbf{x}) = 0$. However, when the objectives are continuous, constructing the indicators from an existing dataset for training a CPE is almost surely equivalent to constructing them in a held-out sense as there will be no ties. Furthermore, if there is observation noise it is beneficial to allow sampling at $\mathbf{x}$ again for noise reduction (e.g. averaging), as the true $\mathbf{f_\bullet}(\mathbf{x})$ may be dominated even if a single observation $\mathbf{y}$ at $\mathbf{x}$ suggests otherwise.

## 3 Incorporating User Preferences

In multi-objective optimization (MOO), practitioners invoke *subjective preferences* to single out a subset of designs to meet application-specific requirements. Ideally, we would like not only to

incorporate these subjective preferences but also to avoid retraining our active generation framework every time a new preference is given.

A standard approach to incorporating subjective preferences is *scalarization*: e.g. for convex scalarization we specify a weight vector $\boldsymbol{\lambda} \in \mathbb{R}^L$ with $\|\boldsymbol{\lambda}\|_1 = 1$ and maximize

$$\underset{\mathbf{x}}{\operatorname{argmax}}\, s_{\boldsymbol{\lambda},\mathbf{r}}(\mathbf{x}), \quad \text{where} \quad s_{\boldsymbol{\lambda},\mathbf{r}}(\mathbf{x}) = \boldsymbol{\lambda}^\top (\mathbf{f_\bullet}(\mathbf{x}) - \mathbf{r}) \tag{10}$$

where $\mathbf{r} \in \mathbb{R}^L$ is a reference point [27, 45, 45, 31]. While scalarization blends objectives according to explicit trade-off weights, it is not optimal when learning a conditional generative model of the Pareto set via a CPE on labels $z_n = \mathbb{1}[s_{\boldsymbol{\lambda},\mathbf{r}}(\mathbf{x}_n) > \tau]$ for some threshold $\tau \in \mathbb{R}$, as in [13], since each new $\boldsymbol{\lambda}$ would require retraining. Furthermore, thresholding weighted objectives can blur the non-dominance boundary compared to directly labeling it.

### 3.1 Preference direction vectors and alignment indicators

As we will see in section 4, our solution to incorporating user preferences for active generation is based on amortization. In other words, instead of estimating a model $q_\phi(\mathbf{x})$ as in VSD, we will learn a conditional model of the form $q_\phi(\mathbf{x}|\mathbf{u})$. Consequently, instead of scalarization, we introduce *preference direction vectors* $\mathbf{u} \in \mathcal{U}$ where $\mathcal{U} = \{\mathbf{u} \in \mathbb{R}^L : \|\mathbf{u}\|_2 = 1\}$, defined from observed or desired (subjective) user specified outcomes. In our experiments, we train our method using

$$\mathbf{u}_n = g(\mathbf{y}_n) := \frac{\mathbf{y}_n - \mathbf{r}}{\|\mathbf{y}_n - \mathbf{r}\|_2}. \tag{11}$$

These unit vectors capture the relative emphasis among objectives in a single geometric object. Given a trained model, a user can specify their own preferences via $\mathbf{u}_\star = g(\mathbf{y}_\star)$ and our approach will generate solutions from $q_\phi(\mathbf{x}|\mathbf{u}_\star)$. Importantly, our generative model needs to enforce that generated samples respect a user's desired trade-off. Therefore, we define an *alignment indicator*, $a \in \{0, 1\}$, that labels each $(\mathbf{x}, \mathbf{u})$ pair as 'aligned' if it achieves correct projection onto the preference direction. We will make clear the need for this indicator variable in the next section.

Preference directions generalize (convex) scalarization weights to a (non-convex) generative setting: any $\boldsymbol{\lambda}$ can be mapped to a unit-norm vector $\mathbf{u} = \boldsymbol{\lambda}/\|\boldsymbol{\lambda}\|_2$, and conversely each $\mathbf{u}$ induces a unique normalized weight. In fact [8, 9] use "reference vectors", which are similar to our $\mathbf{u}$, for scalarization of objectives and to promote diversity for many-objective evolutionary algorithms (MOEAs). By conditioning on $(\mathbf{u}, a)$ rather than scalarizing by $\boldsymbol{\lambda}$, our generative Pareto-set model becomes both more flexible (no retraining for new trade-offs) and more faithful to non-dominance structure. We visualize these preference direction vectors in Figure 1.

## 4 Amortized Active Generation of Pareto Sets

We now have all the components to describe our amortized active generation framework that learns to generate (approximate) solutions in the Pareto set, conditioned on user preferences. We call our method active generation of Pareto sets (A-GPS), and it begins by generalizing the active generation objective in [39]. That is, for each round, $t$, we minimize the reverse KL divergence between the generative model $q_\phi(\mathbf{x}|\mathbf{u})$ and an underlying (unobserved) true model $p(\mathbf{x}|\mathbf{u}, z, a)$,

$$\phi_t^* = \underset{\phi}{\operatorname{argmin}}\, \mathbb{D}_{\mathrm{KL}}[q_\phi(\mathbf{x}|\mathbf{u})p(\mathbf{u}|z)\|p(\mathbf{x}|\mathbf{u}, z, a)p(\mathbf{u}|z)],$$

$$= \underset{\phi}{\operatorname{argmin}}\, \mathbb{E}_{p(\mathbf{u}|z)}[\mathbb{D}_{\mathrm{KL}}[q_\phi(\mathbf{x}|\mathbf{u})\|p(\mathbf{x}|\mathbf{u}, z, a)]]. \tag{12}$$

Here we are actually conditioning on $z = 1$ and $a = 1$, however we leave this implicit henceforth to avoid notational clutter. The inclusion of $p(\mathbf{u}|z)$ rewards learning an *amortized* generative model, $q_\phi(\mathbf{x}|\mathbf{u})$, over the distribution of the relevant preference directions. Naturally we cannot evaluate $p(\mathbf{x}|\mathbf{u}, z, a)$ directly, and so we appeal to Bayes' rule,

$$p(\mathbf{x}|\mathbf{u}, z, a) = \frac{1}{Z}p(z|\mathbf{x}, \mathbf{u})p(a|\mathbf{x}, \mathbf{u})p(\mathbf{x}|\mathbf{u}). \tag{13}$$

Here we have assumed conditional independence between $z$ and $a$ given $\mathbf{x}$ and $\mathbf{u}$, and since $Z = p(z, a|\mathbf{u})$ is a constant w.r.t. $\mathbf{x}$, we will omit it from our objective. We make a further simplifying

assumption that a-priori $p(\mathbf{x}|\mathbf{u}) = p(\mathbf{x}|\mathcal{D}_0)$, and then we rely on the likelihood guidance terms, $p(z|\mathbf{x},\mathbf{u})p(a|\mathbf{x},\mathbf{u})$, to capture the joint relationship between $(\mathbf{x},\mathbf{u})$ in the variational posterior. We justify this decision by noting that the *alignment* relationship, $\mathbf{x}|\mathbf{u}$, may be difficult to reason about a-priori, and requiring such a prior would then preclude the use of pre-trained models for $p(\mathbf{x}|\mathcal{D}_0)$. Putting this all together results in the following equivalent amortized ELBO objective,

$$\phi_t^* = \underset{\phi}{\mathrm{argmax}}\, \mathcal{L}_{\text{A-ELBO}}(\phi) \quad \text{where,} \tag{14}$$

$$\mathcal{L}_{\text{A-ELBO}}(\phi) = \mathbb{E}_{p(\mathbf{u}|z)}[\mathcal{L}_{\text{ELBO}}(\phi)],$$
$$= \mathbb{E}_{\underbrace{p(\mathbf{u}|z)}_{\text{Direction dist.}}}\Big[\mathbb{E}_{q_\phi(\mathbf{x}|\mathbf{u})}\big[\log\underbrace{p(z|\mathbf{x},\mathbf{u})}_{\text{Pareto CPE}} + \log\underbrace{p(a|\mathbf{x},\mathbf{u})}_{\text{Align. CPE}} - \beta\mathbb{D}_{\text{KL}}[q_\phi(\mathbf{x}|\mathbf{u})\|p(\mathbf{x}|\mathcal{D}_0)]\big]\Big]. \tag{15}$$

The $\beta$ coefficient appears here to control the objective's exploration-exploitation tradeoff, and where $\beta = 1$ results in the exact minimization of Equation 12. We will now discuss how we estimate each of these components in turn, leading to the A-GPS algorithm presented in Algorithm 1.

## 4.1 Estimating A-GPS's component distributions

**Preference direction distribution,** $p(\mathbf{u}|z)$**.** Since we observe $\mathbf{u}_n$, we can approximate empirically $p(\mathbf{u}|z) \approx (\sum_{n=1}^N z_n)^{-1}\sum_{n=1}^N z_n \mathbb{1}[\mathbf{u} = \mathbf{u}_n]$. Alternatively, we can use maximum likelihood to learn a parameterized estimator $q_\gamma(\mathbf{u}) \approx p(\mathbf{u}|z)$, with data $\mathcal{D}_N^z = \{(\mathbf{x}_n,\mathbf{u}_n,z_n)\}_{n=1}^N$,

$$\gamma_t^* = \underset{\gamma}{\mathrm{argmin}}\, \mathcal{L}_{\text{Pref}}(\gamma,\mathcal{D}_N^z), \quad \text{where} \quad \mathcal{L}_{\text{Pref}}(\gamma,\mathcal{D}_N^z) = -\frac{1}{\sum_{n=1}^N z_n}\sum_{n=1}^N z_n\log q_\gamma(\mathbf{u}_n). \tag{16}$$

We find this occasionally aids exploration. Examples of appropriate parametric forms are von Mises-Fisher distributions, power spherical distributions [14] or normalizing flows [35]. We find Normal distributions, or mixtures, normalized to the unit sphere are more numerically stable than some of the specialized spherical distributions, see Sec. C.1 for more detail. We also find that fitting an unconditional $q_\gamma(\mathbf{u}) \approx p(\mathbf{u})$ for just the initial round can aid exploration of the Pareto front.

**Pareto CPE,** $p(z|\mathbf{x},\mathbf{u})$**.** As per the original VSD, we define a CPE to directly discriminate over the solution set, $\mathcal{S}$. Sometimes we find setting $\mathcal{S} = \mathcal{S}_{\text{Pareto}}^t$ leads to overly exploitative behavior. Instead, we anneal the set using the Pareto ranking method described in [15], and define the labels based on a thresholded rank, $k$: $z_n = \mathbb{1}[\mathbf{x}_n \in \{\bigcup_k \mathcal{S}_{\text{Pareto},k}^t : \forall k \le \tau_t\}]$. Here $k = 1$ indicates the Pareto set, $k = 2$ the next non-dominated set once $\mathcal{S}_{\text{Pareto},1}^t$ is removed, etc. We use $\tau_t = f_\tau(\{\mathbf{y} : \mathbf{y} \in \mathcal{D}_N\}, \delta_t)$ as presented in [39], ensuring $\tau_T$ labels just $\mathcal{S}_{\text{Pareto}}^t$. The annealed method (Equation 20) applied to quantiles of $-k$ is particularly effective. So, with this label, we use the log-loss to train a CPE,

$$\mathcal{L}_{\text{CPE}}^z(\theta,\mathcal{D}_N^z) = -\frac{1}{N}\sum_{n=1}^N z_n\log\pi_\theta^z(\mathbf{x}_n,\mathbf{u}_n) + (1-z_n)\log(1-\pi_\theta^z(\mathbf{x}_n,\mathbf{u}_n)), \tag{17}$$

where $\pi_\theta^z(\mathbf{x},\mathbf{u})$ is a discriminative model parameterized by $\theta$, e.g. a neural network.

**Preference alignment CPE,** $p(a|\mathbf{x},\mathbf{u})$**.** Since we do not wish to rely on a strong prior, $p(\mathbf{x}|\mathbf{u})$, for our sole-source of preference alignment information, we instead explicitly reward alignment in our conditional generative model by using a CPE guide. We create contrastive data for training this guide, $\mathcal{D}_N^a = \{(a_n=1,\mathbf{x}_n,\mathbf{u}_n)\}_{n=1}^N \cup \bigcup_{i=1}^P\{(a_n=0,\mathbf{x}_n,\mathbf{u}_{\rho_i(n)})\}_{n=1}^N$ where the second set are purposefully misaligned by permutations, $\rho_i : \mathbb{N} \to \mathbb{N}$. This results in the log-loss,

$$\mathcal{L}_{\text{CPE}}^a(\psi,\mathcal{D}_N^a) = -\frac{1}{N+PN}\left[\sum_{n=1}^N\log\pi_\psi^a(\mathbf{x}_n,\mathbf{u}_n) + \sum_{i=1}^P\sum_{n=1}^N\log(1-\pi_\psi^a(\mathbf{x}_n,\mathbf{u}_{\rho_i(n)}))\right], \tag{18}$$

where $\pi_\psi^a(\mathbf{x},\mathbf{u})$ is our CPE parameterized by $\psi$. We make use of two permutation methods for creating the contrastive data. The first is to just use random permutation without allowing any random alignments. The second is to use the top-$k$ nearest neighbors, based on $\mathbf{u}_n$ cosine distance, for $k$ replicates. The random permutation contrastive data covers the space of misalignment, while the top-$k$ permutations improves the angular precision of the alignment scoring CPE. For all experiments we use 7 random permutation replicates, and 2 top-2 replicates, for a total of $P = 9$.

**Algorithm 1** A-GPS optimization loop. See Figure 5 for a visual representation.

---

**Require:** Initial dataset $\mathcal{D}_N$, black-box $\mathbf{f}_\bullet$, prior $p(\mathbf{x}|\mathcal{D}_0)$, CPEs $\pi^z_\theta(\mathbf{x}, \mathbf{u})$ and $\pi^a_\psi(\mathbf{x}, \mathbf{u})$, variational families $q_\gamma(\mathbf{u})$ and $q_\phi(\mathbf{x}|\mathbf{u})$, threshold function $f_\tau$ and $\delta_1$, budget $T$ and $B$.
1: **function** FITMODELS($\mathcal{D}_N, \tau$)
2:      $\mathcal{D}^z_N \leftarrow \{(z_n, \mathbf{x}_n, \mathbf{u}_n)\}^N_{n=1}$, where $z_n = \mathbb{1}[\mathbf{x}_n \in \{\bigcup_k \mathcal{S}^t_{\text{Pareto},k} : \forall k \leq \tau\}]$, $\mathbf{u}_n = (\mathbf{y}_n - \mathbf{r})/\|\mathbf{y}_n - \mathbf{r}\|$
3:      $\mathcal{D}^a_N \leftarrow \{(a_n=1, \mathbf{x}_n, \mathbf{u}_n)\}^N_{n=1} \cup \bigcup^P_{i=1}\{(a_n=0, \mathbf{x}_n, \mathbf{u}_{\rho_i(n)})\}^N_{n=1}$
4:      $\gamma^* \leftarrow \operatorname{argmin}_\gamma \mathcal{L}_{\text{Pref}}(\gamma, \mathcal{D}^z_N)$
5:      $\theta^* \leftarrow \operatorname{argmin}_\theta \mathcal{L}^z_{\text{CPE}}(\theta, \mathcal{D}^z_N)$
6:      $\psi^* \leftarrow \operatorname{argmin}_\psi \mathcal{L}^a_{\text{CPE}}(\psi, \mathcal{D}^a_N)$
7:      $\phi^* \leftarrow \operatorname{argmax}_\phi \mathcal{L}_{\text{A-ELBO}}(\phi, \theta^*, \psi^*, \gamma^*)$
8:      **return** $\phi^*, \theta^*, \psi^*, \gamma^*$
9: **for** round $t \in \{1, \ldots, T\}$ **do**
10:      $\tau_t \leftarrow f_\tau(\{\mathbf{y} : \mathbf{y} \in \mathcal{D}_N\}, \delta_t)$
11:      $\phi^*_t, \theta^*_t, \psi^*_t, \gamma^*_t \leftarrow$ FITMODELS($\mathcal{D}_N, \tau_t$)
12:      $\{\mathbf{u}_{bt}\}^B_{b=1} \leftarrow$ sample $q_{\gamma^*_t}(\mathbf{u})$ or use $\mathbf{u}_\star$
13:      $\{\mathbf{x}_{bt}\}^B_{b=1} \leftarrow$ sample $q_{\phi^*_t}(\mathbf{x}|\mathbf{u}_{bt})$ $\forall b \in \{1, \ldots, B\}$
14:      $\{\mathbf{y}_{bt}\}^B_{b=1} \leftarrow \{\mathbf{f}_\bullet(\mathbf{x}_{bt}) + \boldsymbol{\epsilon}_{bt}\}^B_{b=1}$
15:      $\mathcal{D}_N \leftarrow \mathcal{D}_N \cup \{(\mathbf{x}_{bt}, \mathbf{y}_{bt})\}^B_{b=1}$
16: **return** $\mathcal{D}_N, \phi^*_T, \theta^*_T, \psi^*_T, \gamma^*_T$

---

## 4.2 Learning A-GPS's variational distribution

To learn $q_\phi(\mathbf{x}|\mathbf{u})$, we can now re-write our amortized ELBO, Equation 15, in terms of these estimated quantities,

$$\mathcal{L}_{\text{A-ELBO}}(\phi, \theta, \psi, \gamma) = \mathbb{E}_{q_\gamma(\mathbf{u})}\left[\mathbb{E}_{q_\phi(\mathbf{x}|\mathbf{u})}\left[\log \pi^z_\theta(\mathbf{x}, \mathbf{u}) + \log \pi^a_\psi(\mathbf{x}, \mathbf{u})\right] - \beta \mathbb{D}_{\text{KL}}[q_\phi(\mathbf{x}|\mathbf{u})\|p(\mathbf{x}|\mathcal{D}_0)]\right].$$
(19)

We find that using 'on-policy' gradient estimation methods such as REINFORCE [41, 30] are very slow when we have complex variational distribution forms, $q_\phi(\mathbf{x}|\mathbf{u})$, e.g. causal transformers. This is because we have to set a low learning rate to avoid the variance of this estimator inducing exploding gradients for long sequences. Also, new samples have to be drawn from the variational distribution every iteration of stochastic gradient descent (SGD), which can be computationally expensive. So instead we use an 'off-policy' gradient estimator with importance weights to emulate the on-policy estimator,

$$\nabla_\phi \mathcal{L}_{\text{A-ELBO}}(\phi, \theta, \psi, \gamma)$$
$$= \mathbb{E}_{q_{\phi'}(\mathbf{x}|\mathbf{u})q_\gamma(\mathbf{u})}\left[w(\mathbf{x}, \mathbf{u}) \cdot \left(\log \pi^z_\theta(\mathbf{x}, \mathbf{u}) + \log \pi^a_\psi(\mathbf{x}, \mathbf{u}) - \beta \log \frac{q_\phi(\mathbf{x}|\mathbf{u})}{p(\mathbf{x}|\mathcal{D}_0)}\right) \nabla_\phi \log q_\phi(\mathbf{x}|\mathbf{u})\right].$$
(20)

Here $w(\mathbf{x}, \mathbf{u}) = q_\phi(\mathbf{x}|\mathbf{u})/q_{\phi'}(\mathbf{x}|\mathbf{u})$ are the importance weights [34, 7]. Now we use $S$ samples from $\mathbf{x}^{(s)} \sim q_{\phi'}(\mathbf{x}|\mathbf{u}^{(s)})$, to approximate the expectation in Equation 20. If we choose $\phi' = \phi$ we recover on-policy gradients, however we typically only update $\phi'$ every 100 iterations of optimising A-ELBO, or if the effective sample size drops below a predetermined threshold ($0.33S$). Whenever we update $\phi'$ we also resample $S$ samples. We use these estimated gradients with an appropriate SGD algorithm, such as Adam [25], to optimize for $\phi^*_t$.

## 4.3 Generating Pareto set candidates for evaluation

To recommend candidates for black-box evaluation in round $t$, we sample a set of $B$ designs from our search distribution,

$$\{\mathbf{x}_{bt}\}^B_{b=1} \sim \prod^B_{b=1} q_{\phi^*_t}(\mathbf{x}|\mathbf{u}_{bt}), \quad \text{where} \quad \phi^*_t = \operatorname*{argmax}_\phi \mathcal{L}_{\text{A-ELBO}}(\phi, \theta^*_t, \psi^*_t, \gamma^*_t).$$
(21)

We are free to choose $\mathbf{u}_{bt}$ based on preferences ($\mathbf{u}_\star$); or if we do not have specific preferences to incorporate into the query, we sample $\{\mathbf{u}_{bt}\}^B_{b=1} \sim \prod^B_{b=1} q_{\gamma^*_t}(\mathbf{u})$ for broad Pareto front exploration.

Table 1: Comparison of recent MOG and related techniques. '✓' means the method has the feature, '✗' the method lacks the feature and '–' the method can be easily extended to incorporate the feature. 'Modular' refers to the non-specific nature of the variational distribution used by conditioning by adaptive sampling (CbAS), VSD and A-GPS, i.e., it can be chosen based on the task.

| Method | Designed for MOO | Online black-box optimization (BBO) | Amortized preference conditioning | Non-convex Pareto front | Discrete/mixed $\mathcal{X}$ | Generative pref. model, $q_\gamma(\mathbf{u})$ | Generative obs. model, $q_\phi(\boldsymbol{x})$ | Guide |
|---|---|---|---|---|---|---|---|---|
| LaMBO [38] | ✓ | ✓ | ✗ | ✓ | ✓ | ✗ | Masked LM | nEHVI |
| LaMBO-2 [21] | ✓ | ✓ | ✗ | ✓ | ✓ | ✗ | Diffusion | nEHVI |
| Pareto Set Learning (PSL) [28] | ✓ | ✓ | ✓ | ✓ | ✗ | ✗ | Deterministic MLP | Scalarization |
| GFlowNets [24] | ✓ | ✓ | ✓ | ✓ | ✓ | ✗ | GFlowNets | Scalarization |
| ParetoFlow [44] | ✓ | ✗ | ✗ | ✗ | ✓ | – | Diffusion | Scalarization |
| PROUD [43] | ✓ | ✗ | ✗ | ✓ | – | ✗ | Diffusion | Multiple grad. desc. |
| Preference Guided Diffusion [3] | ✓ | ✗ | ✗ | ✓ | ✗ | ✗ | Diffusion | Preference CPE |
| CbAS [6] | – | – | ✗ | ✓ | ✓ | ✗ | Modular | Dominance CPE |
| VSD [39] | – | ✓ | ✗ | ✓ | ✓ | ✗ | Modular | Dominance CPE |
| A-GPS (ours) | ✓ | ✓ | ✓ | ✓ | ✓ | ✓ | Modular | Dominance CPE |

# 5 Related Work

Our work sits at the intersection of online black-box optimization, generative modeling, and user-guided multi-objective search. We organize existing methods along three dimensions: whether they operate online or offline, whether they directly optimize acquisition functions or learn conditional generative models for optimization, if they use inference-time guidance or learning for generation.

**Online vs. offline.** Traditional MOBO methods, such as hypervolume-based acquisition (EHVI, noisy expected hypervolume improvement (nEHVI), and their variants), entropy search [42, 10, 11, 22] and scalarization methods [27, 45, 31, 13], operate online by sequentially querying the black-box using acquisition rules that balance exploration and exploitation. In contrast, offline MOG approaches like ParetoFlow and guided diffusion frameworks [44, 43, 3] train generative models from a fixed dataset of evaluated designs, without further oracle queries. While these offline methods can leverage rich generative priors, they have not been designed to adapt to new information.

**Generative models vs. acquisition optimization.** Recent advances in "active generation" recast black-box optimization as fitting conditional generative models to high-value regions, guided by predictors and/or acquisition functions. Methods like VSD, GFlowNets, and diffusion-based solvers [39, 24, 17, 21] show that generative search can match or exceed traditional direct acquisition function optimization, particularly in large search spaces. However, existing generative frameworks often need re-training to integrate subjective preferences. Similarly, all direct acquisition optimization methods require additional optimization runs to incorporate new preferences. An exception is Pareto set learning [28], which learns a neural-net, that maps from scalarization weights to designs.

**Guidance vs. learning.** Or inference-time vs. re-training/fine-tuning based search. Guided generation methods, such as those based on guided diffusion and flow matching [21, 43, 44, 3] use a pre-trained generative model, from which samples are then *guided* at inference time such that they are generated from a conditional generative model, leaving the original generative model unchanged. It has been noted in [26] that guided methods, though computationally efficient, may be prone to co-variate shift preventing them being guided too far from the support of the pretrained model. Conversely, learning-based methods such as [39, 40] explicitly re-train or fine-tune the generative model to condition it, thereby circumventing these co-variate shift issues at the cost of more computation, but allowing for less constrained exploration in online scenarios.

Our A-GPS approach unifies these dimensions: it learns an amortized conditional generative model online, bypasses explicit acquisition optimization, and uses sequential learning to avoid co-variate shift. Table 1 compares key features across representative MOG methods.

# 6 Experiments

We now evaluate A-GPS on a number of benchmarks and compare it to some popular baselines. Firstly we apply A-GPS to a number of well known *continuous* synthetic MOO test functions as a proof-of-concept. Then we apply it to three high-dimensional sequence design challenges —

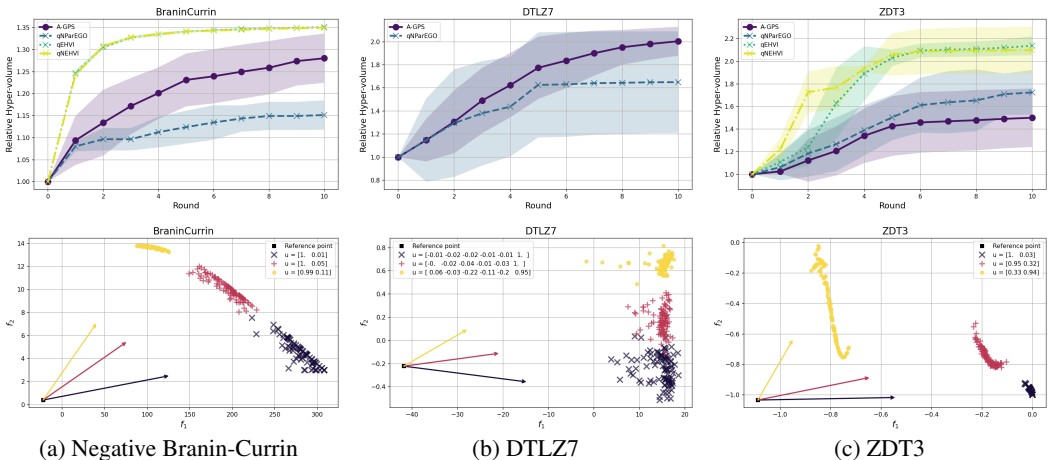

Figure 2: Experimental results on three test functions commonly used in the MOBO literature. The top row reports HVI per round, the bottom row demonstrates amortized preference conditioning by generating Pareto front samples (DTLZ7 is a PCA projection of the front).

A-GPS's intended application — that emulate real protein engineering tasks. Our primary measure of performance is Pareto front relative HVI [46, 19] using the implementation in [4]. For all experiments we set $\beta = 0.5$ as the full KL regularization in Equation 15 can hamper exploitation in later rounds on some tasks. We refer the reader to Appendix D for full experimental details.

## 6.1 Synthetic test functions

As a proof-of-concept, we demonstrate A-GPS on some classical continuous paramterized MOO problems ($\mathbf{x} \in \mathbb{R}^D$) commonly used for MOBO [45, 5, 4]. Even though A-GPS has not been designed for purely continuous problems, they none-the-less allow us to demonstrate some appealing properties of A-GPS. We present three here; negative **Branin-Currin** ($D = 2$, $L = 2$), **DTLZ7** ($D = 7$, $L = 6$), and **ZDT3** ($D = 4$, $L = 2$), see [16, 45, 5]. More detailed descriptions of these functions, additional experimental detail and on additional test functions are presented in Sec. D.1. We apply A-GPS to these continuous domains using a conditional Gaussian generative model, $q_\phi(\mathbf{x}|\mathbf{u}) = \mathcal{N}(\mathbf{x}|\boldsymbol{\mu}(\mathbf{u}), \boldsymbol{\sigma}^2(\mathbf{u}))$, with mean and variance parameterized by a neural network (NN). The top row of Figure 2 reports mean relative hypervolume versus optimization round. The bands indicating $\pm 1$ std. from 10 runs with random parameter initialisation. We compare to three Gaussian process (GP)-based baselines, qNEHVI [11], qEHVI and qNParEGO [10]. All methods use 64 training points, and then recommend $B = 5$ candidates for $T = 10$ rounds, and A-GPS has $\tau_0$ set using the $p = 0.25$ percentile of Pareto ranks. On Branin-Currin, A-GPS (purple) rapidly outpaces the scalarization based qNParEGO, but is dominated by the methods that explicitly estimate EHVI. However, these methods do not scale to the higher dimensional DTLZ7 problem, where A-GPS still outperforms qNParEGO. While A-GPS can model the complex Pareto front of ZDT3, the direct optimization methods outperform it, we suspect a stronger generative backbone for continuous data would help here. The bottom row of Figure 2 illustrates preference conditioning: each panel plots the sampled Pareto front (dots) from $q_\phi(\mathbf{x}|\mathbf{u}_\star)$ colored by three representative preference directions $\mathbf{u}_\star$. These $\mathbf{u}_\star$ were chosen by,

$$\mathbf{y}_\star \in \{[Q_{\mathcal{F}_{\text{Pareto}}^{t,1}}(0.9), Q_{\mathcal{F}_{\text{Pareto}}^{t,2}}(0.1)], [\text{Av}(\mathcal{F}_{\text{Pareto}}^{t,1}), \text{Av}(\mathcal{F}_{\text{Pareto}}^{t,2})], [Q_{\mathcal{F}_{\text{Pareto}}^{t,1}}(0.1), Q_{\mathcal{F}_{\text{Pareto}}^{t,2}}(0.9)]\}, \quad (22)$$

where $Q$ is an empirical quantile function and Av denotes the set mean of each dimension, $l$, of the observed Pareto front, $\mathcal{F}_{\text{Pareto}}^{t,l}$. We then use Equation 11 to convert these into $\mathbf{u}_\star$ with an automatically inferred reference point $\mathbf{r}$. We project the higher dimensional DTLZ7 outcomes into two dimensions using principal component analysis (PCA) for this visualisation. Overall, these results show that A-GPS supports flexible, a-posteriori preference conditioning across a variety of continuous landscapes.

## 6.2 Ehrlich vs. naturalness

We now evaluate A-GPS on a challenging two-objective synthetic 'peptide' design task that couples the Ehrlich synthetic landscape [37] with a ProtBert [18] 'naturalness' score. The Ehrlich function

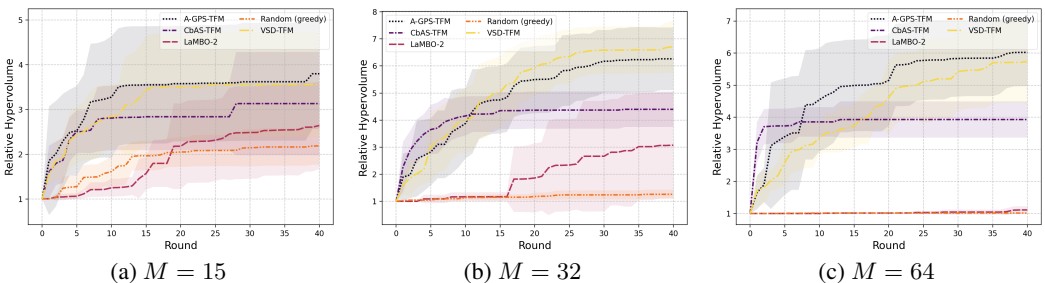

$$\text{(a) } M = 15 \qquad\qquad \text{(b) } M = 32 \qquad\qquad \text{(c) } M = 64$$

Figure 3: Ehrlich function vs. ProtBert naturalness score for different sequence lengths.

has been designed to emulate key aspects of protein fitness; it maps each discrete sequence to a scalar by embedding combinatorial motif interactions in a highly rugged, multi-modal, but artificial terrain. In stark contrast, ProtBert's loss reflects genuine amino-acid patterns learned from 217 million real proteins. We use protein sequences $\mathcal{X} = \mathcal{V}^M$ where $|\mathcal{V}| = 20$ and $M \in \{15, 32, 64\}$. The two objectives are,

$$f_\bullet^1(\mathbf{x}) = \text{Ehrlich}(\mathbf{x}), \qquad f_\bullet^2(\mathbf{x}) = e^{-\mathcal{L}_{\text{ProtBert}}(\mathbf{x})}. \tag{23}$$

Here we convert ProtBert's log-loss, $\mathcal{L}_{\text{ProtBert}}(\mathbf{x})$ into a likelihood ($f_\bullet^2 \in [0, 1]$), making it output in a comparable range to the Ehrlich function ($f_\bullet^1 \in \{-1\} \cup [0, 1]$). We compare against a baseline that randomly mutates the set of best candidates, CbAS [6] and VSD [39], which use the same 2-layer CNN Pareto CPE as A-GPS, and against the guided diffusion based LaMBO-2 [21], which is formulated for discrete MOBO tasks using EHVI guidance. A-GPS uses the same CNN architecture for its alignment CPE. CbAS and VSD use a causal transformer architecture for their $q_\phi(\mathbf{x})$. A-GPS uses the same backbone transformer, but embeds $\mathbf{u}$ for its prefix token and uses FiLM [32] on the transformer embeddings for conditioning the transformer on preferences, $q_\phi(\mathbf{x}|\mathbf{u})$. The same (unconditional) architectures are used as priors by CbAS, VSD and A-GPS, which are trained on the initial sequences using maximum likelihood. Unlike in [39, 6], we allow CbAS to resample $q_\phi(\mathbf{x})$ between rounds (once every 100 iterations) — this drastically improves its performance. Results are reported in Figure 3 for $T = 40$ rounds from random starting conditions (bands indicating $\pm 1$ std). All methods are given 128 training samples, and then recommend batches of size $B = 32$ per round, and $\tau_0$ is set as $p = 0.25$ percentile of Pareto ranks. We use the poli and poli-baselines libraries for running the benchmarks and LaMBO-2 baseline [20]. Additional experimental details and timing results are in Sec. D.2. A-GPS performs similarly to VSD, CbAS tends to be overly exploitative, and LaMBO-2's guided masked-diffusion model tends to under-perform on this task.

### 6.3 Bi-grams

For this experiment we use the bi-grams optimization task from [38] where the aim is to maximize the occurance of three bi-grams ('AV', 'VC' and 'CA') in an $M = 32$ length sequence. We start with 512 random sequences that have no more than three of these bi-grams present. We then have $T = 64$ rounds of $B = 16$ to optimize the bi-gram occurances. The initial Pareto front is sparse so $\tau_0$ is the $p = 0.5$ percentile of Pareto ranks. The same models as the previous experiment are used along with a masked-transformer model (mTFM) backbone for CbAS, VSD and A-GPS. This allows control over the number of mutations to apply to an existing sequence, rather than generating a complete sequence from scratch — see Appendix C for details. LaMBO-2's masked diffusion backbone also has this ability, and following [38] we use a 1-mutation budget for these models. Relative hypervolume improvement results are presented in Figure 4 (a) and (b), sequence diversity is computed as the average pair-wise edit distance between all sequences. We can see the causal transformer backbone A-GPS and VSD models perform best, followed by LaMBO-2. CbAS overfits early as we can see from the diversity plot, and the mTFM models perform similarly to the random baseline and take many rounds to show any improvement on this task.

### 6.4 Stability vs. SASA

Our final experiment uses the simulation-based protein stability vs. solvent accessible surface area (SASA) task from [38]. The aim is to optimize six base red fluorescent proteins with $M > 200$ for stability ($-\Delta G$) and SASA. We use the FoldX black-box implementation in poli [20], with

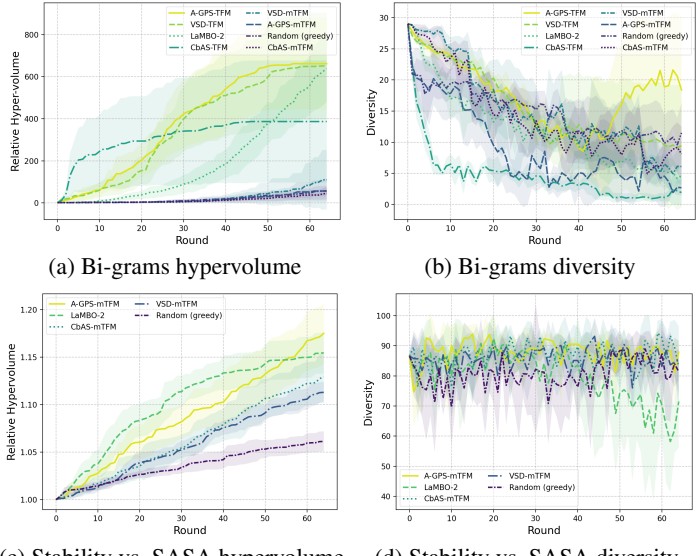

(a) Bi-grams hypervolume  (b) Bi-grams diversity

(c) Stability vs. SASA hypervolume  (d) Stability vs. SASA diversity

Figure 4: Bi-grams and Stability vs. SASA results for relative hypervolume (HV) improvement and diversity.

512 training samples, $T = 64$, $B = 16$, and a budget of one mutation per round. We have a rich starting Pareto front so $\tau_0$ is set from $p = 0.1$. We only use the mTFM backbone for CbAS, VSD and A-GPS as FoldX is best modelling only small differences to the original sequences. The results are summarised in Figure 4 (c) and (d). LaMBO-2 initially performs well, but is overcome by A-GPS as the sequence diversity diminishes. See Sec. D.4 for additional plots of the estimated Pareto front.

# 7 Limitations and Discussion

We have considered the problem of active multi-objective generation, which frames discrete black-box multi-objective optimization as an online sequential generative learning task. Our proposed solution leverages recent advances in generative models to estimate a distribution, $q_\phi(\mathbf{x}|\mathbf{u})$, of the Pareto set directly, conditioned on user preferences $\mathbf{u}$, which we call active generation of Pareto sets (A-GPS).

**Limitations.** A limitation with A-GPS, and one that it shares with many MOBO and MOG methods, is that it can be hard to specify algorithm hyper-parameters a-priori — before new data has been acquired — and the settings of these hyper-parameters can effect real-world performance. We are mindful of this in our implementation and design of A-GPS, and as such it comprises components that can be independently trained and validated meaningfully on the initial training data at hand. In particular, we find A-GPS, VSD and CbAS are sensitive to the prior model used, $p(\mathbf{x}|\mathcal{D}_0)$. To aid practitioners use these methods, we outline a general procedure we find works well in Sec. B.2 for prior choice and initialization. Another limitation with A-GPS as presented in this work is the choice of the generative model for the continuous synthetic test functions. The MLP used does not appear to scale well to higher-dimensional problems, and we expect using flow-matching [29] or diffusion [23] backbones would remedy this issue, and incorporating these models into the VSD and A-GPS frameworks is an active research direction.

Another avenue of future work would be to explore the rich literature on MOEAs for alternative multi-objective measures beyond scalarizations and hypervolume indicators that may be suitable for guided MOG [1]. Additionally, this literature gives a thorough treatment of stochastic experiments in the multi-objective setting [36] which may be useful for MOG.

In contrast to other approaches that are dependent on diffusion models, our choice of generative model is flexible and modular. Our empirical experiments demonstrate that our method performs well on high dimensional sequence design tasks. We hope that our modular framework will result in many future extensions to different architectures for generative models, resulting in further practical algorithms for active generation in large search spaces. For code implementing A-GPS, VSD and all of the experimental results, please see github.com/csiro-funml/variationalsearch.

## Acknowledgements

This work is funded by the CSIRO Science Digital and Advanced Engineering Biology Future Science Platforms, and was supported by resources and expertise provided by CSIRO IMT Scientific Computing. We would like to thank the anonymous reviewers for the constructive feedback and advice, and Sebastian Rojas Gonzalez for pointing us to the literature on MOEAs and for the discussion on estimating PHVI — all of which greatly increased the quality of this work.

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

## A    Broader Impacts

This work is motivated by applications that aim to improve societal sustainability, for example, through the engineering of enzymes to help control harmful waste. However, as with many technologies, it carries the risk of misuse by malicious actors. We, the authors, explicitly disavow and do not condone such uses.

## B    Additional Methodology Details

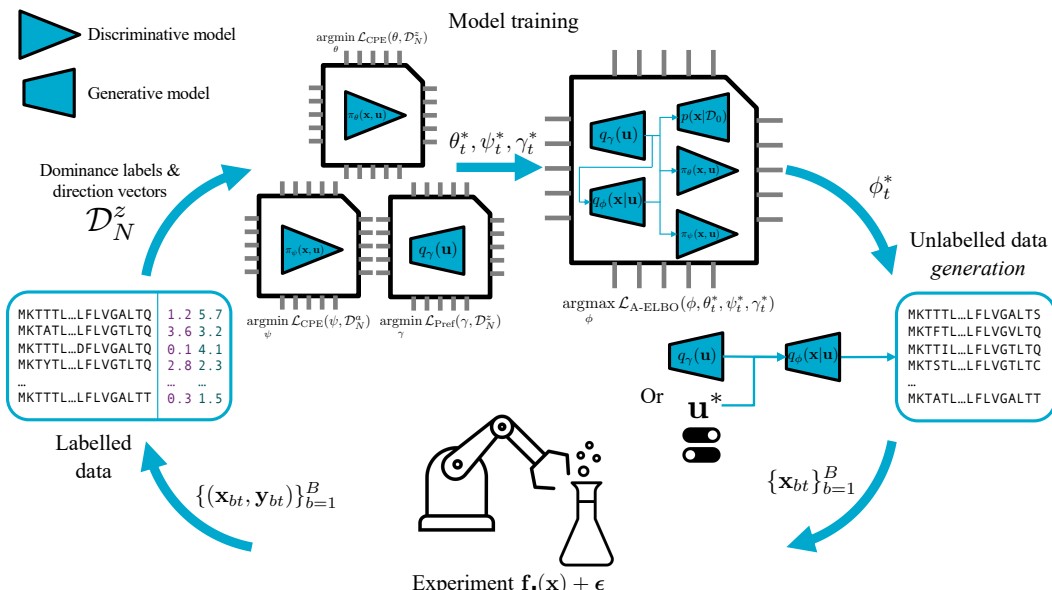

Figure 5: A visual depiction of Algorithm 1 — A-GPS learns all the distributions involved by optimizing different components of a reverse KL loss. At time $t$, the optimized variational distribution $q_\phi(\mathbf{x}|\mathbf{u})$ with parameters $\phi_t^*$ is used to generate new designs that can incorporate new user's preferences $\mathbf{u}^*$. We iterate until a convergence/user criterion is satisfied.

### B.1    A Non-dominance CPE is estimating probability of hypervolume improvement

Using a CPE trained on Pareto non-dominance labels, $z$, is equivalent to estimating the PHVI under some general conditions. The proof is based on the current observed Pareto set and front, $\mathcal{S}_{\text{Pareto}}^t$, $\mathcal{F}_{\text{Pareto}}^t$, respectively, and relies on the box-decomposition definition of hypervolume. We use $\mathbf{y}$ as a shorthand for $\mathbf{y}(\mathbf{x}) = \mathbf{f}_\bullet(\mathbf{x}) + \epsilon$, and $\mathbf{y}'$ for $\mathbf{y}(\mathbf{x}') = \mathbf{f}_\bullet(\mathbf{x}') + \epsilon'$, etc., where the context is clear. Firstly, our proof relies on the following assumptions,

**Assumption 1** (Measurement noise). *No objectives exhibit measurement noise, $\epsilon = \mathbf{0}$.*

This is so our dominance comparison, $\succ$, and hypervolume computation are exact with no ambiguity introduced from noisy measurements. We will discuss the consequences of relaxing this assumption later.

**Assumption 2** (Reference Point). *The reference point, $\mathbf{r} \in \mathbb{R}^L$ is strictly dominated by every feasible objective vector,*

$$\forall \mathbf{y} \in \mathcal{Y} \subset \mathbb{R}^L, \qquad \mathbf{y} \succ \mathbf{r}. \tag{24}$$

This avoids negative-volume boxes when computing hypervolume, i.e. boxes must have positive or no contribution to hypervolume.

We define a box of hypervolume for any $\mathbf{y} \in \mathcal{Y}$ under Assumption 1 and Assumption 2 as,

$$\mathrm{B}(\mathbf{y}) := [\mathbf{r}, \mathbf{y}] = [r_1, y_1] \times [r_2, y_2] \times \cdots \times [r_L, y_L]. \tag{25}$$

Then let $\lambda_L$ denote an $L$-dimensional Lebesgue measure, so the (dominated) hypervolume of a finite set, $\mathcal{A} \subset \mathbb{R}^L$ is,

$$\mathrm{HV}(\mathcal{A}) := \lambda_L \Big( \bigcup_{\mathbf{y} \in \mathcal{A}} \mathrm{B}(\mathbf{y}) \Big). \tag{26}$$

With this we can define hypervolume improvement (HVI),

$$\mathrm{HVI}(\mathbf{x}) := \mathrm{HV}(\mathcal{F}^t_{\mathrm{Pareto}} \cup \{\mathbf{y}(\mathbf{x})\}) - \mathrm{HV}(\mathcal{F}^t_{\mathrm{Pareto}}). \tag{27}$$

In addition, for any $\mathcal{S} \subset \mathcal{X}$, let $\mathrm{Pareto}(\mathcal{S})$ denote the Pareto subset of $\mathcal{S}$, i.e.,

$$\mathrm{Pareto}(\mathcal{S}) := \{\mathbf{x} \in \mathcal{S} : \mathbf{x}' \not\succ \mathbf{x}, \forall \mathbf{x}' \in \mathcal{S} \backslash \mathbf{x}\}. \tag{28}$$

Note that $\mathrm{Pareto}(\mathcal{S} \cup \{\mathbf{x}\}) = \mathrm{Pareto}(\mathcal{S})$ for any $\mathbf{x} \in \mathcal{X}$ that is dominated by an element of $\mathcal{S}$. With these simple definitions and assumptions, and noting that:

$$z(\mathbf{x}) = \mathbb{1}[\mathbf{x} \in \mathrm{Pareto}(\mathcal{S}^t_{\mathrm{Pareto}} \cup \{\mathbf{x}\})], \tag{29}$$

we have the following result.

**Theorem 1** (Equivalence of Indicators)**.** *For every $\mathbf{x} \notin \mathcal{S}^t_{Pareto}$, the HVI indicator is equivalent to a non-dominance indicator,*

$$\mathbb{1}[\mathrm{HVI}(\mathbf{x}) > 0] = z(\mathbf{x}). \tag{8}$$

*Proof.* This is straightforward to see if we consider the dominated and non-dominated cases.

**Case 1: $\mathbf{x}$ is dominated.** Having $z(\mathbf{x}) = 0$ implies that the condition $\mathbf{x}' \not\succ \mathbf{x}, \forall \mathbf{x}' \in \mathcal{S}^t_{\mathrm{Pareto}}$ is not satisfied and the Pareto set remains unchanged, i.e., $\mathrm{Pareto}(\mathcal{S}^t_{\mathrm{Pareto}} \cup \{\mathbf{x}\}) = \mathcal{S}^t_{\mathrm{Pareto}}$, so that $\mathrm{HV}(\mathcal{F}^t_{\mathrm{Pareto}} \cup \{\mathbf{y}\}) = \mathrm{HV}(\mathcal{F}^t_{\mathrm{Pareto}})$. Hence, there is no hypervolume improvement,

$$z(\mathbf{x}) = 0 \quad \Rightarrow \quad \mathbb{1}[\mathrm{HVI}(\mathbf{x}) > 0] = 0. \tag{30}$$

**Case 2: $\mathbf{x}$ is non-dominated.** If $\mathbf{x}' \not\succ \mathbf{x}$ for all $\mathbf{x}' \in \mathcal{S}^t_{\mathrm{Pareto}}$, there is *no* $\mathbf{y}' \in \mathcal{F}^t_{\mathrm{Pareto}}$ such that $\mathrm{B}(\mathbf{y}) \subset \mathrm{B}(\mathbf{y}')$. For $\mathrm{HVI}(\mathbf{x})$ to be positive, we need to show that $\mathrm{HV}(\mathcal{F}^t_{\mathrm{Pareto}}) < \mathrm{HV}(\mathcal{F}^t_{\mathrm{Pareto}} \cup \{\mathbf{y}\})$, i.e., whatever remains of the difference between $\mathrm{B}(\mathbf{y})$ and the previous union $\bigcup_{\mathbf{y}' \in \mathcal{F}^t_{\mathrm{Pareto}}} \mathrm{B}(\mathbf{y}')$ must be a set of positive Lebesgue measure. Indeed, letting $\gamma := \min_{\mathbf{x}' \in \mathcal{S}^t_{\mathrm{Pareto}}} \max_{i \in \{1,...,L\}} y_i - y_i'$, which is positive due to dominance, and setting $\delta := \min\{\gamma/2, \min_{i \in \{1,...,L\}} (y_i - r_i)/2\} > 0$ (by [Assumption 2](#)), we have that the $\delta$-box $\mathrm{B}_\delta := \prod_{i=1}^L [y_i - \delta, y_i] \subset \mathrm{B}(\mathbf{y})$ is not covered by the union $\bigcup_{\mathbf{y}' \in \mathcal{F}^t_{\mathrm{Pareto}}} \mathrm{B}(\mathbf{y}')$. It then follows that $\mathrm{HVI}(\mathbf{x}) = \mathrm{HV}(\mathcal{F}^t_{\mathrm{Pareto}} \cup \{\mathbf{y}\}) - \mathrm{HV}(\mathcal{F}^t_{\mathrm{Pareto}}) \geq \lambda_L(\mathrm{B}_\delta) > 0$, confirming that

$$z(\mathbf{x}) = 1 \quad \Rightarrow \quad \mathbb{1}[\mathrm{HVI}(\mathbf{x}) > 0] = 1. \tag{31}$$

Thus, for any choice $\mathbf{x} \notin \mathcal{S}^t_{\mathrm{Pareto}}$ we have $z(\mathbf{x}) = \mathbb{1}[\mathrm{HVI}(\mathbf{x}') > 0]$. $\square$

**Corollary 1** (Non-Dominance CPE estimates PHVI)**.** *Following straightforwardly from [Theorem 1](#),*

$$\mathbb{P}(z(\mathbf{x}) = 1|\mathbf{x}) = \mathbb{P}(\mathrm{HVI}(\mathbf{x}) > 0|\mathbf{x}) =: \mathrm{PHVI}(\mathbf{x}), \quad \forall \mathbf{x} \notin \mathcal{S}^t_{Pareto}, \tag{9}$$

*as the events are equivalent. Thus, a CPE trained on $z$, using a proper loss, is predicting PHVI.* $\square$

It is worth noting that we do not require [Assumption 1](#) for these indicators to be equivalent so long as the same noisy samples, $\mathbf{y}_i$, are being used when computing them. However, we can no longer claim that the indicators themselves are reliably computing hypervolume improvement or non-dominance since the addition of noise makes the underlying comparisons ambiguous with respect to the true objectives, $\mathbf{f}_\cdot$. It may still be possible to show convergence of A-GPS to the true Pareto set, $\mathcal{S}_{\mathrm{Pareto}}$, while using noisy observations. The authors of VSD [39] were able to guarantee convergence using a CPE trained with noisy PI labels for single objective BBO using results from the neural tangent kernel (NTK) literature (see their Appendix F). Some of these results may be extended to Pareto set classification, though we leave this for future work, as all of our experiments are noise-free.

### B.2 Fitting the prior

We found that A-GPS, VSD and CbAS, which share generative backbones, were all very sensitive to the choice of prior. For complex problems, the best results were obtained in general when the prior is chosen to be of the same form (or an unconditional variant) as the variational distribution, and fit to the $T = 0$ training data. However, for flexible backbone models like transformers, it can be easy to overfit to this training data. We found that adding dropout in conjunction with an early stopping training procedure on the initial training dataset reliably led to good performance for all methods. The exact procedure used is listed as follows,

1. Set prior dropout $p$

2. Train prior with a 10% validation set, make note of number of iterations when validation loss begins to increase

3. Train prior with all data for the number of iterations noted in the previous step

4. If appropriate, copy weights to variational distribution (without dropout).

When the prior and variational models were compatible, we initialized the variational model with these learned prior weights. We run an ablation of this procedure in Sec. E.3.

## C Architectural Details

### C.1 Preference direction distributions

In all the experiments we use a mixture of isotropic Normal distributions where the samples have been constrained to the unit norm,

$$q_\gamma(\mathbf{u}) = \sum_{k=1}^{K} w_k \mathcal{N}_{\|\mathbf{u}\|}(\mathbf{u}|\boldsymbol{\mu}_k, \boldsymbol{\sigma}_k^2), \tag{32}$$

and $\gamma = \{(\boldsymbol{\mu}_k, \boldsymbol{\sigma}_k)\}_{k=1}^{K}$. Typically, we find $K = 5$ is sufficient. We learn this via maximum likelihood as per Equation 16, but we add an extra regularisation term: $-\frac{1}{K} \sum_{k=1}^{K} (\|\boldsymbol{\mu}_k\| - 1)^2$ so the magnitude of the mixture means is controlled (and does not decrease to $0$ or increase to $\pm\infty$). We have compared this to von Mises distributions, and find it more numerically stable, we also find no tangible benefit using more complex spherical normalizing flow representations [35]. Furthermore, we find that the performance is similar to, if not slightly superior to, the empirical approximation,

$$q_\gamma(\mathbf{u}) = \frac{1}{\sum_{n=1}^{N} z_n} \sum_{n=1}^{N} z_n \mathbb{1}[\mathbf{u} = \mathbf{u}_n], \tag{33}$$

where $\gamma = \{\mathbf{u}_n : z_n = 1\}_{n=1}^{N}$. Though on occasions when only a few observations define the Pareto front, we find that using this representation can lead to an overly exploitative strategy.

### C.2 Sequence variational distributions

In this section we summarize the main variational distribution architectures considered for A-GPS VSD and CbAS.

**Causal Transformer.** For some of the sequence experiments we implemented an auto-regressive (causal) transformer of the form,

$$q_\phi(\mathbf{x}) = \text{Categ}(x_1|\text{softmax}(\phi_1)) \prod_{m=2}^{M} q_{\phi_d}(x_m|x_{1:m-1}), \quad \text{where}$$

$$q_{\phi_{1:m}}(x_m|x_{1:m-1}) = \text{Categ}(x_m|\text{softmax}(\text{DTransformer}_{\phi_d}(x_{1:m-1}))). \tag{34}$$

For details on the decoder-transformer with a causal mask see [33, Algorithm 10 & Algorithm 14] for maximum likelihood training and sampling implementation details respectively.

**Masked Transformer.** We also implemented a masked transformer model (mTFM) that learns to mutate an initial sequence, $\mathbf{x}'$, at a set of positions, $\mathbf{o} = [o_1, \ldots, o_M]$ where $o_m \in \{0, 1\}$ and $\sum_{m=1}^M o_m = O$ for a mutation budget, $O \in \{1, \ldots, M\}$. The complete generative model is,

$$q_\phi(\mathbf{x}, \mathbf{o}|\mathbf{x}') = q_{\phi_e}(\mathbf{x}|\mathbf{x}'[\mathbf{o} \leftarrow x_{\text{mask}}])q_{\phi_{oe}}(\mathbf{o}|\mathbf{x}'). \tag{35}$$

Where the notation $\mathbf{x}'[\mathbf{o} \leftarrow x_{\text{mask}}]$ means we apply a masking token to the original sequence at the positions indicated by $\mathbf{o}$. The mask generation model is,

$$q_{\phi_{oe}}(\mathbf{o}|\mathbf{x}') = \text{Multinomial}(\mathbf{o}|\text{NN}_{\phi_o}(\text{ETransformer}_{\phi_e}(\mathbf{x}'))), \tag{36}$$

Here ETransformer is an encoder-transformer, see [33, Algorithm 9] for details, and $\text{NN}_{\phi_o}$ is a NN decoder, with a convolutional residual layer for capturing additional local structure. The token generation model is,

$$q_{\phi_{xe}}(\mathbf{x}|\mathbf{x}'[\mathbf{o} \leftarrow x_{\text{mask}}]) = \prod_{m=1}^M \begin{cases} \mathbb{1}[x_m = x'_m] & \text{if } o_m = 0; \\ \text{Categ}(x_m|\mathbf{p}_m) & \text{if } o_m = 1, \end{cases} \quad \text{where}$$

$$[\mathbf{p}_1, \ldots, \mathbf{p}_M] = \mathbf{softmax}(\text{ETransformer}_{\phi_e}(\mathbf{x}'[\mathbf{o} \leftarrow x_{\text{mask}}])).$$

Here $\mathbf{softmax} : \mathbb{R}^{|\mathcal{V}| \times M} \to [0, 1]^{|\mathcal{V}| \times M}$. The same encoder-transformer is shared between the mask and token heads, and only the masked positions are allowed to sample new tokens. For our experiments we bias the categorical distribution so that the original token is not resampled. The choice of the set of seed sequences, $\{\mathbf{x}'_i\}_{j=1}^J$, can drastically impact performance of A-GPS, VSD and CbAS. A simple heuristic that we find works well in practice is to uniformly sample with replacement from the current active Pareto set, $\mathcal{S}_{\text{Pareto}}^t$, so $\mathbf{x}' \sim \mathcal{U}(\mathcal{S}_{\text{Pareto}}^t)$. This is similar to the strategy used in LaMBO-2, though since they use EHVI, they can maintain only the top-$B$ sequences per round ranked by EHVI, where $B$ is batch size.

Finally, if we use this model as a prior, we drop the conditional mask model, $q_{\phi_{oe}}(\mathbf{o}|\mathbf{x}')$, and draw mask-positions independently with a Bernoulli distribution, $\text{Bern}(o_m|p_{\text{mask}} = 0.15)$. Then we use [33, Algorithm 12] for training.

We list the configurations of the transformer variational distributions in Table 2. We use additive positional encoding for all of these models. When using these models for priors or initialization of variational distributions, we find that over-fitting can be an issue. To circumvent this, we use dropout and early stopping, see Sec. B.2 for details.

**Conditioning.** As mentioned in the text, for the conditional generative models, $q_\phi(\mathbf{x}|\mathbf{u})$, for A-GPS, we use the same architectures already discussed, but also learn a sequence prefix embedding from $\mathbf{u}$, as well as a simple 1-hidden layer MLPs for implementing FiLM [32] adaptation of the sequence token embeddings,

$$\mathbf{e}_m = \mathbf{e}'_m \circ (1 + f_\alpha(\mathbf{u})) + f_\beta(\mathbf{u}), \quad \text{where} \quad \mathbf{e}_0 = f_{\text{prefix}}(\mathbf{u}). \tag{37}$$

Where $\mathbf{e}'_m$ are the unmodified token embeddings (i.e. outputs of DTransformer or ETransformer), and $\circ$ indicates element-wise product. $f_\alpha$ and $f_\beta$ are the FiLM MLPs, and $f_{\text{prefix}}$ is the prefix embedding MLP input into the transformers (often we find just a linear projection is adequate). We initialize the transformer weights in these models from their non-conditional counterparts when they are used as priors.

Table 2: Transformer network configuration for the sequence experiments.

| ↓ **Property** / $M \to$ | **Ehrlich vs. Nat** | | | **Bi-grams** | **Stability vs. SASA** |
|---|---|---|---|---|---|
| | 15 | 32 | 64 | 32 | > 200 (variable) |
| Layers | 2 | 2 | 2 | 2 | 2 |
| Network Size | 128 | 128 | 128 | 128 | 256 |
| Attention heads | 4 | 4 | 4 | 4 | 4 |
| Embedding size | 64 | 64 | 64 | 64 | 64 |
| FiLM hidden size | 128 | 128 | 128 | 128 | 128 |
| Prior dropout $p$ | 0.5 | 0.4 | 0.2 | 0.2 | 0.1 |
| Mutation budget $O$ | – | – | – | 1 | 1 |

## C.3 Class probability estimator architectures

For all of our experiments we share the same architecture for both $\pi_\theta^z(\mathbf{x}, \mathbf{u})$ and $\pi_\psi^a(\mathbf{x}, \mathbf{u})$. On the continuous synthetic test functions we use the MLP in Figure 6 (a), where we simply concatenate the inputs $\mathbf{x}$ and $\mathbf{u}$. Here `Skip` is a skip connection which implements a residual layer, and `MaxAndAvg` means a scalar weighted sum of max and average pooling.

For the sequence experiments we use the convolutional architecture given in Figure 6 (b). For VSD and CbAS we simply add on another `LayerNorm` and `LeakyReLU` and then an output linear layer. For A-GPS we concatenate $\mathbf{u}$ to output of this CNN, and then pass this concatenation into the MLP in Figure 6 (c). All architectural properties are listed in Table 3.

Table 3: CPE configurations for A-GPS, VSD and CbAS.

| $\downarrow$ **Property** / $M \rightarrow$ | **Ehrlich vs. Nat** | | | **Bi-grams** | **Stability vs. SASA** | **Synthetic Fns.** |
|---|---|---|---|---|---|---|
| | 15 | 32 | 64 | 32 | > 200 (variable) | – |
| E | 16 | 16 | 16 | 16 | 16 | – |
| C | 64 | 64 | 64 | 64 | 96 | – |
| Kc | 5 | 5 | 5 | 5 | 7 | – |
| Kx | 3 | 3 | 3 | 3 | 5 | – |
| Sx | 2 | 2 | 2 | 2 | 4 | – |
| H | 128 | 128 | 128 | 128 | 192 | $\min\{16D, 128\}$ |
| hidden_layers | – | – | – | – | – | 2 |

# D  Experimental Details

## D.1 Synthetic test functions

We tested A-GPS on a number of popular MOO synthetic test functions against strong GP-based baselines. All the test functions are summarised as follows, with additional results presented in Figure 7.

**Branin-Currin** ($D = 2$, $L = 2$): We optimize the negative Branin-Currin convex pair. We found the negative function has a more interesting Pareto front while remaining a challenging MOBO task.

**DTLZ7** ($D = 7$, $L = 6$): A higher-dimensional constraint surface, made of $L$ segments. The outcome dimension is too high for methods that directly estimate the improvement to hypervolume.

**ZDT3** ($D = 4$, $L = 2$): A complex, non-convex front comprised of several disconnected segments, which stresses an optimizer's capacity for both exploration and front-segment coverage.

**DTLZ2** ($D = 3, L = 2$): A smooth, spherical front in the negative orthant, which tests an algorithm's ability to approximate non-convex curved manifolds in higher dimensions.

**DTLZ2** ($D = 5, L = 4$): A higher dimensional instantiation of the DTLZ2 function for testing performance with higher dimensional inputs and objectives.

**GMM** ($D = 2, L = 2$): Here each objective is implemented as a Gaussian mixture model, and so is highly multimodal [12]. This is run without observation noise.

A detailed description of these functions and/or their Pareto-front geometries can be found in [16, 45, 5]. We use BoTorch [4] for the implementations of all the synthetic test functions and the baseline MOBO methods. All experiments were initialized using Latin hyper-cube sampling. The original design space is $\mathcal{X} = [0, 1]^D$ for all problems, and we optimize directly in this space with all methods. The GP based methods use an optimizer that respects these bounds directly, and we clamp A-GPS's generative model to these bounds. We found transforming the space (e.g. using logit-sigmoid transforms) generally led to worse performance.

For A-GPS we use the mixture model in Equation 32 for the preference direction distribution, and for the conditional generative model we use a simple MLP,

$$q_\phi(\mathbf{x}|\mathbf{u}) = \mathcal{N}\big(\mathbf{x}\big|\boldsymbol{\mu}(\mathbf{u}), \boldsymbol{\sigma}^2(\mathbf{u})\big). \tag{38}$$

Here $\boldsymbol{\mu}(\mathbf{u}), \boldsymbol{\sigma}^2(\mathbf{u})$ are MLPs with 2 or 4 hidden layers (if $D \geq 3$) of size of $\min(16D, 256)$ with skip-connections and layer normalization, making them residual networks. We otherwise use the same experimental settings for the rest of the experiments, which are given in Table 4.

```
Sequential(
    Linear(
        in_features=D + L,
        out_features=H
    ),
    LayerNorm(),
    LeakyReLU(),
    Dropout(p=0.1),
    *[Skip(
        Linear(
            in_features=H,
            out_features=H
        ),
        LayerNorm(),
        LeakyReLU(),
        Dropout(p=0.1),
    ) for _ in range(hidden_layers)],
    Linear(
        in_features=H,
        out_features=1
    ),
)
```

(a) Continuous MLP architecture

```
Sequential(
    Linear(
        in_features=H + L,
        out_features=H
    ),
    LeakyReLU(),
    LayerNorm(),
    Skip(
        Linear(
            in_features=H,
            out_features=H
        ),
        LeakyReLU(),
    ),
    Linear(
        in_features=H,
        out_features=1
    ),
```

(c) Sequence-preference concatenation MLP
architecture

```
Sequential(
    Embedding_And_Positional(
        num_embeddings=A,
        embedding_dim=E
    ),
    Dropout(p=0.2),
    Conv1d(
        in_channels=E,
        out_channels=C
        kernel_size=Kc
    ),
    GroupNorm(),
    LeakyReLU(),
    Dropout(p=0.1),
    MaxAndAvgPool1d(
        kernel_size=Kx,
        stride=Sx,
    ),
    Skip(
        Conv1d(
            in_channels=C,
            out_channels=C,
            kernel_size=Kc,
        ),
        GroupNorm(),
        LeakyReLU(),
    )
    AdaptiveMaxAndAvgPool1d(),
    Linear(
        out_features=H
    ),
)
```

(b) Sequence CNN architecture

Figure 6: CPE architectures used for the experiments in PyTorch-like syntax. $\mathtt{A} = |\mathcal{V}|$, $\mathtt{L} = L$ corresponding to $\mathbf{y} \in \mathbb{R}^L$ and $\mathtt{D} = D$ corresponding to $\mathbf{x} \in \mathbb{R}^D$ for the continuous experiments. LaMBO-2 uses the same kernel size as our CNNs. See Table 3 for specific property settings.

## D.2 Ehrlich vs. Naturalness

The Ehrlich vs. naturalness score benchmark was implemented using the `poli` benchmarking library [20], where we implemented our own ProtGPT2-based naturalness black box, and used the inbuilt Ehrlich function [37] (not the holo version). For A-GPS we use the aforementioned generative and discriminative models, otherwise the settings are given in Table 6. We use a modified version of the LaMBO-2 algorithm [21] from `poli-baselines` [20]. We use the following Ehrlich function configurations:

$M = 15$: motif length = 3, no. motifs = 2, quantization = 3

$M = 32$: motif length = 4, no. motifs = 3, quantization = 4

$M = 64$: motif length = 4, no. motifs = 4, quantization = 4

Additional experimental settings are given in Table 6, and runtimes in Table 5.

## D.3 Bi-grams

For the bi-grams experiment we implemented our own black-box for the `poli` library based on the experiment in [38]. All architectural details are presented in Appendix C, with additional experimental settings in Table 6. We also report runtimes in Table 7.

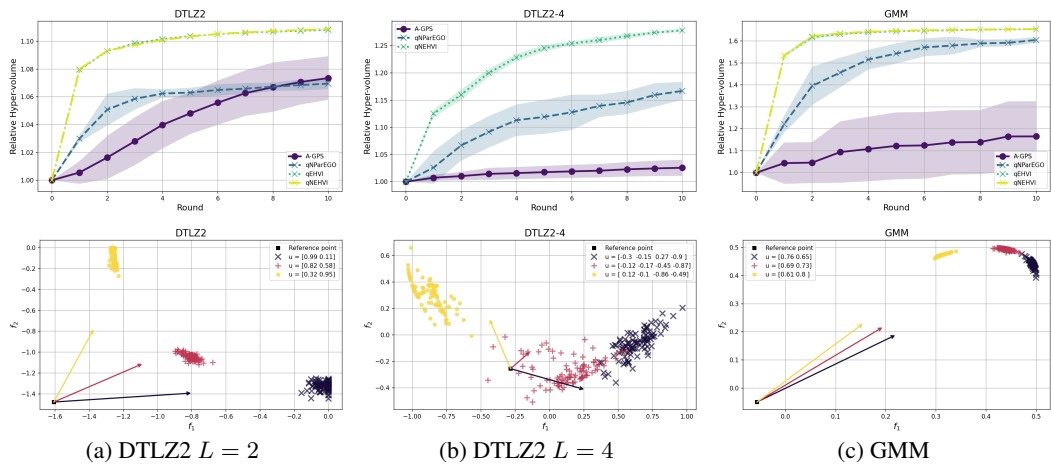

(a) DTLZ2 $L = 2$  (b) DTLZ2 $L = 4$  (c) GMM

Figure 7: Experimental results on an additional three test functions commonly used in the MOBO literature. The top row reports HVI per round, the bottom row demonstrates amortized preference conditioning by generating Pareto front samples (DTLZ2 with $L = 4$ is a PCA projection of the front).

Table 4: Synthetic test functions experimental settings.

| Setting | Value |
|---|---|
| $N_{t=0}$ | 64 |
| $T$ | 10 |
| Replicates | 10 |
| $B$ | 5 |
| $S$ | 256 |
| $\mathbf{r}$ | inferred using BoTorch's `infer_reference_point()` |
| Base BoTorch model | `SingleTaskGP` |
| Optimizer | Adam for A-GPS (lr = $10^{-5}$) and L-BFGS for the GPs |
| Max optimization iter. for A-GPS | 3000 |
| GP restarts | 10 |
| Bounds | $[0, 1]^D$ |
| GP kernel | Matern $\nu = 2.5$ ARD (standardized inputs) |
| GP hyperparameters | LogNormal priors on $\sigma$ and $l$ |

## D.4 Stability vs. SASA

For the stability vs. SASA experiment we used bi-objective black-box from the `poli` library, and the seed sequences and settings from the experiment in [38]. All architectural details are presented in Appendix C, with additional experimental settings in Table 6. We also report runtimes in Table 7, and show some samples from the A-GPS generative model, as well as the empirical Pareto front in Figure 8.

## D.5 Computational resources

All experiments were run on a Dell PowerEdge XE9640 rack server cluster with NVIDIA H100 GPUs and 4th generation Intel Xeon CPUs. All of our models could easily fit on one GPU, and typically took less than 2 hours to complete the experiments.

Table 5: Ehrlich vs. Naturalness times (mins).

| | $M = 15$ | | | $M = 32$ | | | $M = 64$ | | |
| Method | mean | min | max | mean | min | max | mean | min | max |
|---|---|---|---|---|---|---|---|---|---|
| A-GPS-TFM | 18.10 | 17.79 | 18.38 | 19.40 | 18.94 | 19.78 | 21.43 | 20.99 | 21.75 |
| VSD-TFM | 12.43 | 12.26 | 12.61 | 13.62 | 13.10 | 14.34 | 15.85 | 15.24 | 17.26 |
| CbAS-TFM | 9.18 | 8.91 | 9.51 | 9.73 | 9.40 | 9.98 | 11.78 | 11.43 | 12.11 |
| LaMBO-2 | 14.15 | 13.50 | 14.74 | 16.17 | 15.74 | 16.41 | 17.96 | 17.68 | 18.75 |
| Random (greedy) | 0.53 | 0.51 | 0.54 | 0.91 | 0.91 | 0.92 | 2.35 | 2.35 | 2.36 |

Table 6: Sequence experimental settings.

| ↓ Setting / $M \rightarrow$ | Ehrlich vs. Nat. | | | Bi-grams | Stability vs. SASA |
|---|---|---|---|---|---|
| | 15 | 32 | 64 | 32 | > 200 (variable) |
| $N_{t=0}$ | 128 | 128 | 128 | 512 | 512 |
| $T$ | 40 | 40 | 40 | 64 | 64 |
| Replicates | 5 | 5 | 5 | 5 | 5 |
| $B$ | 32 | 32 | 32 | 16 | 16 |
| $S$ | 256 | 256 | 256 | 256 | 256 |
| $\mathbf{r}$ | [-1, 0] | [-1, 0] | [-1, 0] | [0, 0, 0] | auto |
| Threshold $\tau_0$ percentile | 0.25 | 0.25 | 0.25 | 0.5 | 0.1 |

Table 7: Bigrams times (mins).

| Method | mean | min | max |
|---|---|---|---|
| A-GPS-TFM | 30.99 | 30.04 | 31.89 |
| A-GPS-mTFM | 40.32 | 38.78 | 42.19 |
| VSD-TFM | 21.23 | 20.10 | 22.14 |
| VSD-mTFM | 28.13 | 27.93 | 28.44 |
| CbAS-TFM | 14.63 | 14.41 | 14.84 |
| CbAS-mTFM | 24.24 | 23.11 | 26.40 |
| LaMBO-2 | 42.03 | 40.63 | 43.04 |
| Random (greedy) | 0.47 | 0.47 | 0.48 |

# E    Ablation Studies

In this section we test some of the architectural decisions we have made when designing A-GPS.

## E.1    On-policy vs. off-policy gradients

We introduce a new gradient estimator in Equation 20 based on off-policy importance weighting approximations to the on-policy gradient estimator used by [39]. To test its efficacy, we re-run the Ehrlich vs. naturalness score experiments with this new estimator and the original on-policy variant. We report performance and runtimes in Table 9 for $M = 32$. There does not seem to be a consistent difference between the two gradient estimators in terms of hypervolume performance, but runtime is significantly lower for the off-policy estimator, being almost an order of magnitude less for the off-policy variant.

## E.2    Off-policy gradient estimator samples

We test the effect on performance of the number of samples, $S$, used for estimating the gradients of the off-policy estimator, Equation 20, used for A-GPS and VSD in Table 10. We also do the same for the CbAS estimator. We generally find that more samples lead to more performance for all methods, however this effect plateaus for A-GPS and VSD starting at $S = 256$, whereas CbAS still sees improvement beyond $S = 512$. This is also something noted by the original authors in [6].

## E.3    Prior regularization

We found that A-GPS, VSD and CbAS, which share generative backbones, were all very sensitive to the choice of prior. For complex problems best results were obtained in general when the prior is fit to the $T = 0$ training data, however for flexible backbone models like transformers, it can be easy to

Table 8: Stability vs. SASA times (mins).

| Method | mean | min | max |
|---|---|---|---|
| A-GPS-mTFM | 120.75 | 118.52 | 122.67 |
| VSD-mTFM | 108.93 | 107.57 | 109.92 |
| CbAS-mTFM | 97.48 | 96.11 | 99.16 |
| LaMBO-2 | 59.33 | 57.55 | 61.18 |
| Random (greedy) | 11.38 | 10.87 | 12.20 |

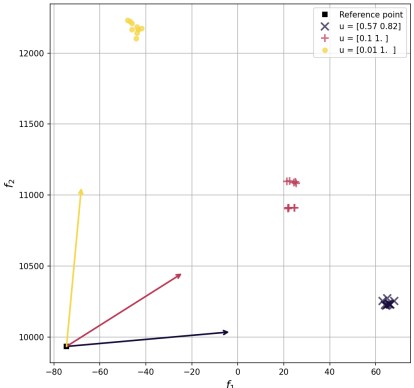
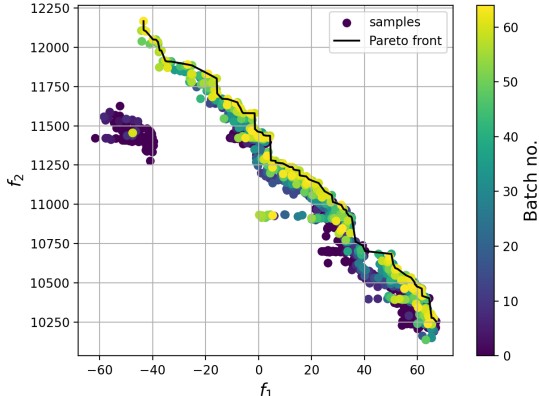

(a) Samples from the final A-GPS generative model conditioned on preference direction vectors.

(b) FoldX black-box evaluations of sequences generated by A-GPS, colored by evaluation round.

Figure 8: Additional A-GPS sample visualisations from the stability vs. SASA experiment in section 6.

Table 9: Ehrlich function vs. naturalness score ablation for $M = 32$. Run time and performance comparison for the on-policy gradient estimators ('-reinf.') vs. the importance weighted off-policy estimators for the A-GPS and VSD methods. All times are in minutes.

| Method | Time (min) | | | $T = 40$ **relative HV improvement** |
|---|---|---|---|---|
| | mean | min | max | |
| A-GPS-TFM | 19.40 | 18.94 | 19.78 | 6.264 (1.159) |
| VSD-TFM | 13.62 | 13.10 | 14.34 | 6.711 (0.952) |
| CbAS-TFM | 9.73 | 9.40 | 9.98 | 4.398 (0.652) |
| LaMBO-2 | 16.17 | 15.74 | 16.41 | 3.074 (1.950) |
| Random (greedy) | 0.91 | 0.91 | 0.92 | 1.260 (0.169) |
| A-GPS-TFM-reinf. | 105.09 | 100.20 | 109.52 | 6.656 (0.920) |
| VSD-TFM-reinf. | 97.95 | 90.74 | 115.81 | 6.257 (1.338) |

overfit. Adding dropout only to the prior generative model is an effective means of controlling this overfitting when used with a validation set to infer the number of learning iterations, as in the process outlined in Sec. B.2.

To demonstrate the effect of over- and under-fitting on performance, we change the transformer dropout probability while holding all else constant, e.g. fitting iterations, on the Ehrlich vs. naturalness experiment for $M = 32$ in Table 11. The training iterations were tuned while setting $p = 0.4$, and so we would expect other values of dropout to be suboptimal, and we can indeed see this is so. All methods severely under-perform with and overfit prior $p \in \{0.2, 0.3\}$. A-GPS and VSD perform slightly worse with an under-fit prior, whereas CbAS's performance actually improves. This is a trend we have noticed in all experiments; prior overfitting leads to poor performance, and under-fitting is less consequential. CbAS tends to favour exploitation, which explains why a broader prior, encouraging exploration, helps the method.

Table 10: Round $T = 40$ relative hypervolume improvement results for the varying the number of samples used to estimate the gradients with the off-policy gradient estimator, Equation 20, and the CbAS gradient estimator.

| $M$ | Method | $S = 64$ | $S = 128$ | $S = 256$ | $S = 512$ |
|---|---|---|---|---|---|
| 32 | A-GPS-TFM | 5.663 (1.357) | 6.212 (0.905) | 6.264 (1.159) | 6.245 (1.398) |
| | VSD-TFM | 5.665 (0.918) | 6.218 (1.041) | 6.711 (0.952) | 6.964 (0.899) |
| | CbAS-TFM | 4.048 (1.249) | 4.167 (0.707) | 4.398 (0.652) | 5.531 (1.347) |
| 64 | A-GPS-TFM | 5.655 (1.646) | 5.522 (1.188) | 6.021 (1.052) | 6.269 (1.808) |
| | VSD-TFM | 5.435 (0.844) | 5.198 (1.240) | 5.738 (1.305) | 4.633 (0.817) |
| | CbAS-TFM | 4.168 (1.352) | 3.803 (0.741) | 3.929 (0.564) | 5.092 (0.972) |

Table 11: Round $T = 40$ relative hypervolume improvement results for the varying the prior dropout probability of the transformer backbone. We only present results for $M = 32$, and the training regime was originally optimized for $p = 0.4$.

| Method | $p = 0.2$ | $p = 0.3$ | $p = 0.4$ | $p = 0.5$ |
|---|---|---|---|---|
| A-GPS-TFM | 3.786 (0.925) | 5.525 (1.224) | 6.264 (1.159) | 5.743 (0.849) |
| VSD-TFM | 3.585 (0.894) | 4.455 (0.433) | 6.711 (0.952) | 6.496 (1.246) |
| CbAS-TFM | 2.543 (0.589) | 4.099 (0.678) | 4.398 (0.652) | 5.838 (1.090) |

### E.4 Empirical vs. parameterized preference direction distribution

For all of our experiments we use the constrained mixture of Normal distributions ($K = 5$), Equation 32, as our parameterized preference directions distribution, $q_\gamma(\mathbf{u})$. We now wish to validate this choice by comparing it to two simpler alternatives: a single constrained Normal distribution ($K = 1$), and the empirical distribution in Equation 33.

We make these comparisons on the synthetic test functions, as their Pareto fronts are well sampled and diverse in their shapes. As can be seen in the results in Figure 9, there is not a consistent leader among the different preference distribution parameterizations across all of these functions. Perhaps the empirical preference distribution strikes the optimal blend of simplicity and performance — however we do occasionally find that it under-performs compared to the parameterized distributions in cases where the Pareto front is sparsely sampled, which happens in some sequence experiments.

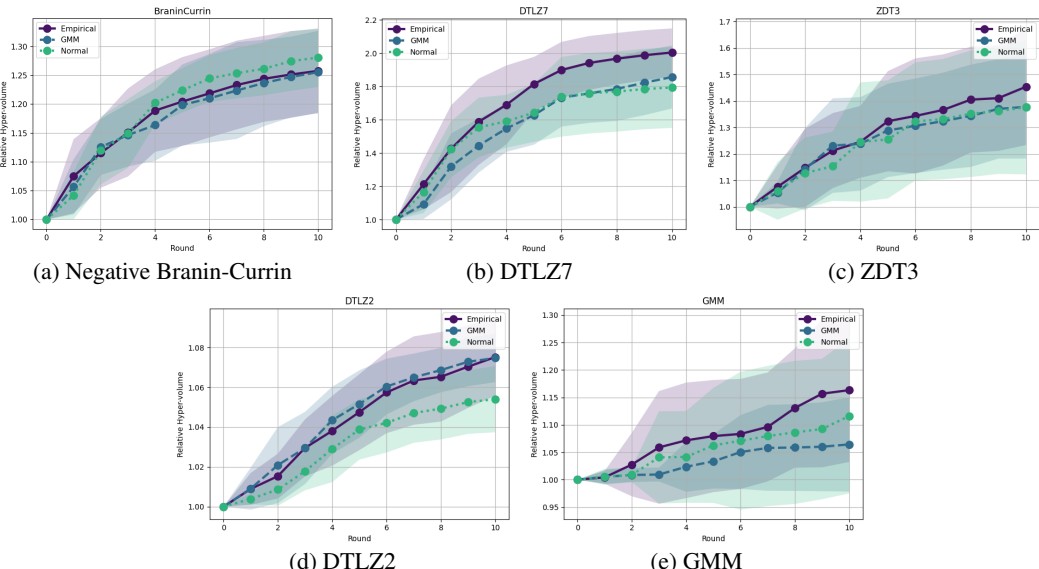

Figure 9: A-GPS preference distribution ablation experimental results on the synthetic test functions used for Sec. 6.1.

