# OpenReview forum: "Amortized Active Generation of Pareto Sets"
_NeurIPS.cc/2025/Conference — NeurIPS 2025 poster_

### Official Review · Reviewer_yJhX · 2025-06-10

**Clarity:** 3
**Significance:** 3
**Originality:** 2
**Rating:** 3
**Confidence:** 3

**Summary:**

The authors propose a method for generating Pareto fronts for black-box multi-objective optimisation tasks. They achieve this by using a generative model that creates samples conditioned on being on the Pareto set, and on alignment with preference vectors set by the user.

The modelling is based on active generation of sets guided by a class probability estimator. Where an estimator is first fit to model the distribution $p(z = 1 | x)$, and then a generative model is created to approximate $p(x | z)$. In this work, user preferences are further incorporated by considering alignment with preference vectors $u$. This results in the need to first approximate three distributions: $p(z | x, u)$ and $p(a | x, u)$ which are modelled using CPEs, and $p(u | z)$ which is modelled with  a von Mises-Fischer distribution and MLE. All the distributions together are then used to fit the ELBO of the generative model using an off-policy gradient estimator with importance weights.

The model is then shown to work well in a variety of real-world benchmarks.

**Questions:**

- Generally see my "weaknesses" section.

- Is there a way a user can set a preference over the spread of the generation in the model?

- Why do you think that your method which does not calculate hyper-volume improvement, beats methods that do go through the explicit calculations and try to maximise it?

**Ethical Concerns:**

["NO or VERY MINOR ethics concerns only"]

**Final Justification:**

Even though the rebuttal made some promising points, I keep my score, as it is difficult to verify if everything in the rebuttal will be added to the paper entirely, so I think the paper would benefit from one final iteration before publication.

**Limitations:**

No, the authors claim to discuss limitations in the related work section, however, these are not spelled out clearly. What the limitations are should be written more clearly.

**Quality:**

2

**Strengths And Weaknesses:**

Strengths:

- Quality: The paper is technically sounds, uses well known techniques and justifies the methods.

- Clarity: The paper is well written, with good figures, equations, and tables. Most of the concepts get across well and can be understood easily.

- Significance: Pareto front generation is of great importance in many scientific disciplines, and this paper proposes a good solution to the problem.

- Originality: The general ideas of the paper rely on previous work of Steinberg et al. (2025) (ICLR), however, there is enough extensions and challenges introduced.

Weaknesses:

- Quality: I don't think the training of alignment vectors is well justified, indeed intuitively if $x_1$ and $x_2$ are close in space, the alignment should reflect stronger than if they were apart. However, by training on the permutations only there is no difference between the two scenarios. This is particularly concerning when we see in the experiment section, that the spread of points conditions on specific preferences can still be wide. Furthermore, the fact that pointing a preference vector ever so slightly outside the Pareto front results in poor suggestions from the algorithm is concerning, and is probably related to this fact.

Furthermore, in the results, it appears that some of the methods have been implemented poorly, e.g. why is CbAS-TFM always outperformed by Random search.

- Clarity: How to set the preference vectors remains a little unclear. I think a clearer example of how a practitioner would go about choosing the vectors and how this would affect the Pareto front would improve the paper. Additionally, I find it confusing about what the paper means by "amortised", since indeed there is re-training of the generative model at each iteration of the optimisation loop which results in a large expensive procedure between iterations -- the only amortisation that I understood related to changing preference vectors however I think this makes the paper premise and title misleading.

Overall, I think the paper could be improved to address the alignment issue, which is both the main strength of the method but appears to have many pitfalls. However, it is mostly technically sound and addresses an important area, so it is in the borderline.

Minor comments:
- In Figure 3, we see the lines change for Random between the first plot and the rest, making it difficult to read
- In Eq. (17) should the D_KL lie outside the inner expectation?

---

> ### Author Rebuttal · Authors · 2025-07-31
>
> Thank you to reviewer yJhX for the challenging questions. We will attempt to answer them below.
>
> **Q1: I don't think the training of alignment vectors is well justified, indeed intuitively if $x_1$ and $x_2$ are close in space, the alignment should reflect stronger than if they were apart ... the fact that pointing a preference vector ever so slightly outside the Pareto front results in poor suggestions from the algorithm is concerning, and is probably related to this fact.**
>
> Thank you for this feedback. One main motivation for using the permutation method for training the alignment CPE was its simplicity --- A-GPS is already complex, and we do not wish to add more hyperparameters if they are not essential. To this end, we have found that increasing the amount of permuted pairs, $(u_{\rho(n)}, x_n)$, to the aligned pairs $(u_n, x_n)$ by about 5:1 drastically improves the performance of the alignment component of A-GPS, and appears to improve the issue seen with the BraninCurrin test function. This procedure is similar to noise contrastive estimation (particularly InfoNCE, that uses negative example pairs), and MaxEnt estimation (where only positive labels exist, and the negative pairs are background points). This procedure rewards an estimator for discriminating relevant examples (aligned pairs) from irrelevant unaligned background/noisy samples.
>
> Furthermore, we were reluctant to make additional smoothness assumptions about the $\mathcal{X} \times \mathcal{U}$ space, since in many applications these may not be valid. For example, one amino acid change in the active site of a protein can drastically impact the protein's properties (e.g. if a hydrophilic amino acid is swapped for a hydrophobic one both structural and functional properties are effected), changing the location of the protein in $\mathcal{Y}$ and hence also $\mathcal{U}$. We can include this point in the paper, and flag this as a direction for future investigation/improvement.
>
> **Q2: It appears that some of the methods have been implemented poorly, e.g. why is CbAS-TFM always outperformed by Random search**
>
> We use the same implementation as [Steinberg, 2025] used in comparison to VSD --- though the authors do note that it is originally designed as an offline method, and it has been adapted to the online setting. However, we did notice the performance of the method improve when we changed the labelling strategy to Pareto rank thresholding, as discussed with reviewer VCzN. This is also more inline with how CbAS is used in both the VSD paper, and in the original implementation --- increasing thresholds. We also found an issue with the LaMBO2 baseline (as implemented in poli-baselines), where the code was not using the correct acquisition function for MOO settings. We have fixed this, and have seen a marked improvement in the performance of the method. However, We still find A-GPS and VSD generally outperform these baselines.
>
> **Q3: How to set the preference vectors remains a little unclear. I think a clearer example of how a practitioner would go about choosing the vectors and how this would affect the Pareto front would improve the paper**
>
> Thank you for the feedback. We will happily clarify this in the paper. Assuming we start with a trained model, $q(x|u)$, perhaps the easiest way for a user to choose a preference vector, $u_\star$, for generation of designs is to start with a desired outcome, $y_\star$. We can then use Equation 9 to normalise this to a preference direction vector, $u_\star$, to input to $x_\star \sim q(x|u=u_\star)$. To control Pareto front exploration, we could parameterise a $q(u)$ directly, rather than learning similarly to that in [25]. We have opted to learn a $q(u)$ in our presentation as we are aiming for maximising the discovered hyper-volume in the experiments.
>
> **Q4: I find it confusing about what the paper means by "amortised", since indeed there is re-training of the generative model at each iteration of the optimisation loop which results in a large expensive procedure between iterations -- the only amortisation that I understood related to changing preference vectors however I think this makes the paper premise and title misleading.**
>
> Amortization in the context of variational inference typically refers to using a parameterized global inference function for the variational distribution, $q$, in contrast to the factored mean-field (independent) approach for each latent variable/parameter. This term came into being to describe variational models like Kingma and Welling's VAE, which can learn conditional distributions, $q(z|x)$, that can be queried with new $x$ without retraining. We can view our variational distribution, $q(x|u)$ in the same way --- we can query $q$ for various $u$ without the need for re-training, as factored mean field (and other MOBO) approaches would require. We can clarify this usage in the text, and perhaps we can re-word line 205 "amortization->retraining" if this was the source of confusion.
>
> **Q5: In Figure 3, we see the lines change for Random between the first plot and the rest, making it difficult to read**
>
> We found keeping the order of the colouring consistent with algorithm performance was clearer for these sorts of plots, but we can re-style if this was confusing, and keep the style aligned with the methods.
>
> **Q6: In Eq. (17) should the $D_{KL}$ lie outside the inner expectation?**
>
> The $D_{KL}$ term has to remain inside the expectation to be consistent with our objective in Equation 10. We can see this in the expansion in Equation 13, and then noting $q(u) \approx p(u|z)$.
>
> **Q7: Is there a way a user can set a preference over the spread of the generation in the model?**
>
> Probably the easiest way to do this post-learning would be to query $q(x|u)$ with multiple $u$, or draw them from a specific distribution --- e.g. bounded uniform over a subset of $\mathcal{U}$. Our framework also supports specifying a $q(u)$ rather than learning one during optimization if that is sensible for a particular application.
>
> **Q8: Why do you think that your method which does not calculate hyper-volume improvement, beats methods that do go through the explicit calculations and try to maximise it?**
>
> This is a very good question, and we believe it is because in quite general circumstances the Pareto non-dominance label _is equivalent to a hyper-volume improvement indicator_ if a box-decomposition method is used. We sketch this now; if we define hyper-volume improvement,
>
> $\text{HVI}(x) = \text{HV}(\mathcal{F} \cup \\{y(x)\\}) - \text{HV}(\mathcal{F})$,
>
> where $\text{HV}$ is hyper-volume, and $\mathcal{F}$ is the Pareto front. We define hyper-volume as a union of the hyper-volume of hyper-rectangles defined by the vertices,
>
> $\text{HV}(\mathcal{F}) = \lambda_L (\bigcup_{y \in \mathcal{F}}[r, y])$,
>
> for a reference point $r \in \mathbb{R}^L$ and a measure, $\lambda_L$. Then the hyper-volume improvement indicator is, $1[\text{HVI}(x) > 0]$. We make two observations,
>
> 1. If $y(x)$ is dominated, then $z = 1[x \in \mathcal{S}] = 0$, but also the hypervolume is unchanged since $\lambda_L([r, y(x)]) \subseteq \text{HV}(\mathcal{F})$, and so $1[\text{HVI}(x) > 0] = 0$.
> 2. If $y(x)$ is not dominated, then $z = 1[x \in \mathcal{S}] = 1$ and also $\lambda_L([r, y(x)]) \not\subseteq \text{HV}(\mathcal{F})$ which means it must contribute positive volume, and so $1[\text{HVI}(x) > 0] = 1$.
>
> This requires the reference point to be dominated by every feasible point, and also that there are no ties. The latter is not true for objectives that are discrete, and so in these circumstances the dominance indicator is not the same as HVI.
>
> Therefore, in the continuous case, with a proper loss (e.g. logistic), training a class probability estimator on the dominance indicator should also estimate the probability of hyper-volume acquisition. This nicely parallels the probability of improvement setting in VSD, and we can extend our paper to present this.

---

> > ### Comment · Reviewer_yJhX · 2025-08-02
> >
> > Thanks to the authors for taking the time to go over my questions, and providing such a detailed response.
> >
> > > Furthermore, we were reluctant to make additional smoothness assumptions about the $\mathcal{X} \times \mathcal{U}$ space, since in many applications these may not be valid. For example, one amino acid change in the active site of a protein can drastically impact the protein's properties (e.g. if a hydrophilic amino acid is swapped for a hydrophobic one both structural and functional properties are effected), changing the location of the protein in $\mathcal{Y}$ and hence also $\mathcal{U}$. We can include this point in the paper, and flag this as a direction for future investigation/improvement.
> >
> > Yes, I think some kind of illustrative example showing smoothness can be detrimental would help justify this.
> >
> > > However, We still find A-GPS and VSD generally outperform these baselines.
> >
> > Thanks for checking this!
> >
> > > Amortization in the context of variational inference typically refers to using a parameterized global inference function for the variational distribution, $q$, in contrast to the factored mean-field (independent) approach for each latent variable/parameter.
> >
> > I agree! I just thought it was the other type of amortisation when I originally read the title, I just think the distinction could be made clearer early on, in order to avoid other people getting confused.
> >
> > > We found keeping the order of the colouring consistent with algorithm performance was clearer for these sorts of plots, but we can re-style if this was confusing, and keep the style aligned with the methods.
> >
> > I much prefer the style aligned with methods, since it allows me to track each method's performance across all benchmarks, instead of having to re-read the legend each time, e.g. I like to see how each method compares to Random. However, feel free to ask for other opinions of course.
> >
> > > This is a very good question, and we believe it is because in quite general circumstances the Pareto non-dominance label is equivalent to a hyper-volume improvement indicator if a box-decomposition method is used.
> >
> > Ah this is an extremely interesting point! I think expanding on this, and adding it to the paper would be very beneficial.

---

> > > ### Author Response · Authors · 2025-08-02
> > > **Thank you**
> > >
> > > Esteemed Reviewer,
> > > \
> > > \
> > > We sincerely thank you for engaging with our rebuttal, and for your valuable comments. Rest assured we will incorporate all your suggestions accordingly. Meantime, kindly let us know if there is anything else we can answer, clarify or improve.
> > > \
> > > \
> > > Best regards,
> > > \
> > > Authors

---

> > > > ### Author Response · Authors · 2025-08-06
> > > >
> > > > Thank you again, yJhX, for your thoughtful feedback.
> > > >
> > > > We would like to double check if you still have any remaining concerns we can clarify while we are still in the discussion phase?
> > > >
> > > > Otherwise, our publication plan for incorporating your feedback is as follows:
> > > >
> > > > 1. Provide an extended justifcation for the alignment training procedure, with smoothness concerns mentioned.
> > > > 2. Improving the baselines CbAS and LaMBO-2. Done, and the results are in the paper, and we will make plotting consistent (we generate plots separately from generating results, so this is trivial)
> > > > 3. Clarify our presentation of amortization (make it clear at the beginning of the document, and clarify line 205).
> > > > 4. We will add our proof of showing Pareto CPE is equivalent to PHVI estimation. This is a straightforward proof, but we think adds nice theoretical justification that was missing.
> > > >
> > > > Thank you again for your help with this work :-)

---

> > > > > ### Comment · Reviewer_yJhX · 2025-08-06
> > > > >
> > > > > Thanks for the responses and clear resolutions. I do think they are a lot of changes for a rebuttal, so I would encourage the authors to be careful about how the paper will be restructured; I think if applied in the best way, the changes will make the paper a much stronger submission.

---

> ### Author Response · Authors · 2025-08-07
>
> Thank you for your concerns, and recognition that we can make this paper "much stronger".
>
> We actually think that the requested changes, from all reviewers, are not major. Most requests are about providing additional experiments or ablations, which we have done, or textual clarifications of our method and its limitations, which exist in these discussions and are easily included in the main text and/or extended discussions in the appendix. The work is, for all intents and purposes, done, and visible here _publicly_ (once this session concludes).
>
> We have not had any requests for major changes to our methodology or approach (and some of the minor requests were actually anticipated by us and already in the code), our novelty has not be challenged by any reviewer, and our core conclusions have remained the same -- or have been actually strengthened, e.g. we can now simply prove that a Pareto CPE is a cheap approximation of PHVI.
>
> We also wish to thank you again and sincerely for your commitment and _continued engagement_, especially as serving as a reviewer through this discussion phase is a lot of work (particularly if you are defending your own work(s) simultaneously).

---

### Official Review · Reviewer_VCzN · 2025-06-22

**Clarity:** 3
**Significance:** 3
**Originality:** 3
**Rating:** 4
**Confidence:** 4

**Summary:**

This paper proposes Amortized Conditional Pareto Estimation (A-CPE), a conditional generative modeling framework for online multi-objective optimization (MOO) that enables preference specification at inference time. The authors develop an autoregressive variational distribution $q_{\phi}(x∣u)$ (motivated by their sequence design tasks) conditioned on preference vectors $u$, which are sampled from an empirical approximation of $p(u∣z)$to amortize over $u$. Pareto-optimality and preference-alignment are modeled using discriminative classifiers $p(z∣x,u)$ and $p(a∣x,u)$, which are integrated into the ELBO objective. To accelerate optimization, the authors employ importance-weighted sampling for gradient estimation and only resample new trajectories when the likelihood ratio under significantly changes. The method is applied online per BO iteration and demonstrates competitive performance to baselines like qEHVI and qNEHVI on bi-objective benchmarks, including both continuous input and discrete sequence design tasks.

**Questions:**

**On the benefit of a generative model**: In low-dimensional continuous input spaces (e.g., Sec. 6.1), what is the practical advantage of using a generative model over directly optimizing $p(z=1|x,u)p(u|z=1)$ via gradient-based or heuristic methods? A controlled comparison against direct optimization would clarify the end-to-end utility of the generative model.

**Ethical Concerns:**

["NO or VERY MINOR ethics concerns only"]

**Final Justification:**

I think regarding the concern about $q(u∣z)$ potentially hindering exploration, the authors have referenced some in-house experiments suggesting that it does not cause significant differences with varying their choice of $z$, and have indicated they will incorporate further analysis in an updated version of the paper. Since new figures cannot be provided at this stage, and based on the overall consistency of the other responses, I choose to accept this explanation and have adjusted my score accordingly.

**Limitations:**

- As noted above, the reliance on an empirical $p(u|z)$ may limit exploration unless preferences are explicitly provided. This should be acknowledged as a limitation or mitigated.
- The scalability of the method on the number of objectives remains unclear. In particular, amortizing over preference vectors  becomes increasingly difficult as the number of objectives grows, a clear elaboration of its scalability boundary, or a dicussion of future work or a limitation claim should be done on this.

**Paper Formatting Concerns:**

The format looks good to me.

**Quality:**

2

**Strengths And Weaknesses:**

Strengths

- Introduces a flexible generative method for conditional Pareto set sampling that allows user preference specification at inference time without retraining.
- The use of an autoregressive sequence model is appropriate for discrete design problems and mitigates the difficulty of searching over combinatorial spaces.
- Avoids explicit hypervolume computation, potentially improving scalability.
- The paper is well-written.

Weakness
- **Key concern** on $q(u|z)$ and random sampling step in Algorithm 1 (lines 11–12) relies on previously observed Pareto-optimal points. In settings where user preferences are not specified and the initial data has poor Pareto Frontier coverage, this empirical distribution will not support sufficient exploration of the Pareto front. This should be either addressed explicitly or the scope of the method should be reframed as targeting preference-aware MOO, rather than general MOO.
- Time and computational cost profiling is missing. A breakdown of time per component (CPE training, generative sampling, ELBO optimization) would help assess scalability.
- While the paper claims to avoid costly hypervolume (HV) computation, in bi-objective settings HV is computationally cheap. An empirical demonstration in higher-dimensional MOO (e.g., 4–5 objectives) would strengthen the claim of improved efficiency over HV-based methods.

I would be happy to reconsider my score if the author could clarify my key concern and the question below.

---

> ### Author Rebuttal · Authors · 2025-07-31
>
> We thank reviewer VCzN for the thoughtful feedback, and will attempt to address their concerns and questions now.
>
> **Q1: Key concern on $q(u|z)$ and random sampling step in Algorithm 1 (lines 11–12) relies on previously observed Pareto-optimal points. In settings where user preferences are not specified and the initial data has poor Pareto Frontier coverage, this empirical distribution will not support sufficient exploration of the Pareto front. This should be either addressed explicitly or the scope of the method should be reframed as targeting preference-aware MOO, rather than general MOO.**
>
> Thank you for this observation. Just as point of clarification, we use a constrained mixture of Gaussians over preferences, rather than an empirical distribution, for all experiments. We also study the consequence of an empirical distribution in Appendix E.2. Even so, after the submission of this version of the paper we were also concerned that a small Pareto set could lead to overly exploitative behaviour. We have subsequently made two small adjustments to A-GPS which we believe rectify this issue.
>
> 1) For the first round, we fit an unconditional $q(u)$ using all available $y$.
> 2) Rather than just using Pareto dominance labels, $z = 1[x \in \mathcal{S}]$, we
>   can use the Pareto ranking method (non-dominance sorting) described in [Deb, 2002], and then specify labels based on a thresholded rank ($s$), $z = 1[x \in \\{\bigcup \mathcal{S}_s : \forall s \leq \tau_t \\}]$. Here $s=1$ indicates the Pareto set, $s=2$ the next non-dominated set (shell) once $\mathcal{S}_1$ is removed, etc. We are then able to choose a threshold, $\tau_t$, per round $t$ using the same mechanisms as the original VSD paper --- and we find the annealed quantile method, Equation 20 of [Steinberg, 2025] (on negative ranks, $-s$), directly applicable. We typically set $\tau_0$ to the 75th-percentile of (negative) ranks, and $\tau_T$ so only the Pareto set is positively labelled.
>
> We do not find much difference in the experimental results with these
> modifications, with the exception of the ZDT3 synthetic test function, in which
> we see an improvement for A-GPS. We report updated round-10 results to reviewer uzmJ.
>
> **Q2: Time and computational cost profiling is missing. A breakdown of time per component (CPE training, generative sampling, ELBO optimization) would help assess scalability.**
>
> We have reported overall run-times for the methods in the supplementary material, Appendix E.1. We will expand upon these though for all seqeunce experiments. For example, here is the run-time
> summary (in minutes) of the additional methods for the Ehrlich vs. Naturalness experiments, for M = 64, as well as VSD and A-GPS (which are in the Appendix)
>
> | Method | mean | min | max |
> |-|-|-|-|
> A-GPS-LSTM | 6.66 | 5.70 | 8.54
> A-GPS-TFM | 9.52 | 7.51 | 11.49
> VSD-LSTM | 6.11 | 4.63 | 7.62
> VSD-TFM | 7.97 | 7.11 | 8.27
> CbAS-LSTM | 3.56 | 3.35 | 3.72
> CbAS-TFM | 5.20 | 5.08 | 5.40
> LaMBO-2 | 2.32 | 2.30 | 2.33
> Random (greedy) | 0.07 | 0.07 | 0.07
>
> The runtime of the CPEs depends on many factors, such as their architecture,
> training epochs etc, but typically we find them to be less than 10% of the
> total runtime --- which is dominated by the A-ELBO computation. These timings
> above, and those in Appendix E.1 should hopefully provide insight as to the
> contribution of the generative model sampling in the runtime. We can include
> this discussion in the paper. The main determinant of scalability is the expense of sampling the generative model. For a transformer this is quadratic in sequence length.
>
> **Q3: While the paper claims to avoid costly hypervolume (HV) computation, in bi-objective settings HV is computationally cheap. An empirical demonstration in higher-dimensional MOO (e.g., 4–5 objectives) would strengthen the claim of improved efficiency over HV-based methods.**
>
> This is an excellent point, and we will include additional results summarised below for round 10 on two higher dimensional problems,
>
> | Test function ($L$) | Method | Mean round 10 relative HVI (1 std.) |
> |-|-|-|
> |DTLZ7 ($L=6$)| A-GPS | 3.073 (1.363) |
> | | qParEGO | 2.313 (1.205) |
> |DTLZ2 ($L=5$)| A-GPS | 1.053 (0.022) |
> | | qParEGO | 1.042 (0.027) |
>
> Evaluating these problems with EHVI-based is at or beyond their limits (it is claimed qNEHVI can scale to $L=5$, but we could not run the method in time for this rebuttal). We hope this helps to highlight the efficacy of our method.
>
> **Q4: On the benefit of a generative model: In low-dimensional continuous input spaces (e.g., Sec. 6.1), what is the practical advantage of using a generative model over directly optimizing $p(z=1 | x, u)p(u|z=1)$ via gradient-based or heuristic methods? A controlled comparison against direct optimization would clarify the end-to-end utility of the generative model.**
>
> This is an interesting question, however, note that
>
> $\text{argmax}_x p(z=1 | x, u)p(u|z=1)$
>
> Will optimize $x$ independently of $p(u|z=1)$, potentially ignoring any preferences. Perhaps a better alternative would be,
>
> $\text{argmax}\_x \mathbb{E}_{p(u|z=1)}[p(z=1 | x, u)p(a=1 | x, u)]$,
>
> where the expectation could be approximated with samples (corresponding to batch size). Or for a particular preference $u_\star$,
>
> $\text{argmax}\_x p(z=1 | x, u_\star) p(a=1 | x, u_\star)$.
>
> This will not yield an _amortized_ model for preferences, and each new preference, $u_\star$, _will require a separate optimization_. In low dimensional spaces, this may not be costly, in which case this is probably less direct and effective than scalarization methods like MBORE [10] or qParEGO [8]. We expect the benefit of A-GPS in the, low dimensional, continuous case comes from its stochastic nature, perhaps analogously to stochastic vs. deterministic optimization algorithms providing some robustness to local optima. If the reviewer still feels such a study is of benefit, this (modified) direct optimization is simple to implement as an ablation.
>
> Our focus is really on high-dimensional and discrete optimization problems, where direct optimization is impossible without e.g. latent space optimization or reparameterisation via a generative model and the utility of our method is clear --- we will clarify this in the text.
>
> **Q5: The scalability of the method on the number of objectives remains unclear. In particular, amortizing over preference vectors becomes increasingly difficult as the number of objectives grows, a clear elaboration of its scalability boundary, or a dicussion of future work or a limitation claim should be done on this.**
>
> As can be seen be the DTLZ7 ($L=6$) and DTLZ2 ($L=5$) experiments the method scales more effectively than some of its EHVI-based competitors --- the main cost being the non-dominance checking, which is worst case $\mathcal{O}(LN^2)$, or more typically $\mathcal{O}(LN \log N)$. We also visualised a (PCA) projection of the preference vectors and sampled points, like that in Figure 2, and found nice separated clustering of samples for each input vector. Sadly we are unable to share the figures, but these will be in the updated version of the manuscript, along with the hyper-volume improvement results.
>
> ---
>
> [Deb, 2002] K. Deb, A. Pratap, S. Agarwal and T. Meyarivan, "A fast and elitist multiobjective genetic algorithm: NSGA-II," in IEEE Transactions on Evolutionary Computation, vol. 6, no. 2, pp. 182-197, April 2002, doi: 10.1109/4235.996017.

---

> > ### Comment · Reviewer_VCzN · 2025-08-04
> >
> > Thank the authors for the detailed rebuttal. Some additional minor comments: for Q1 and Q4, it would be helpful to include the elaboration on the modeling of $q(u∣z)$ for different values of $z$, as mentioned above, as part of an ablation study. Optionally, regarding Q5, my view — and it would be helpful if the authors could comment if they already have an impression on this — is that the algorithm likely exhibits a performance sweet spot, outperforming exact scalarization-based maximization methods at certain objective dimensionalities. However, as the number of objectives increases, amortized preference learning may become increasingly challenging and potentially less accurate than scalarization-based methods. Since the method performs competitively with qParEGO on the DTLZ2/7 benchmarks, discussing this dimensionality–performance trade-off explicitly would strengthen the claim. If strong performance holds or improves in even higher-dimensional settings, that would further reinforce the method’s claim.

---

> > > ### Author Response · Authors · 2025-08-05
> > >
> > > Thank you to reviewer VCzN for the remarks.
> > >
> > > **For Q1 and Q4, it would be helpful to include the elaboration on the modeling of $q(u∣z)$ for different values of $z$, as mentioned above, as part of an ablation study**
> > >
> > > Of course, we are thinking the following ablations will be insightful, and will begin working on them now:
> > >
> > > 1. The effect of fitting unconditional $q(u)$ for round $t=1$, and then the effect of different $\tau_0$ Pareto shell thresholds. Perhaps the Ehrlich vs. Naturalness experiment will be a good experiment for this ablation, as it has the worst starting Pareto front coverage.
> > > 2. The ablation study mentioned in Q4 above.
> > >
> > > **However, as the number of objectives increases, amortized preference learning may become increasingly challenging and potentially less accurate than scalarization-based methods. Since the method performs competitively with qParEGO on the DTLZ2/7 benchmarks, discussing this dimensionality–performance trade-off explicitly would strengthen the claim.**
> > >
> > > This is a really interesting discussion point, and it also relates to our response to Q4 above. In circumstances where scalarization can be used, scalarization will almost surely be more accurate in respecting trade-off preferences compared to sampling from our conditional generative model --- irrespective of dimensionality of $y$. However, each new selection of preferences requires a *new scalarization weighting, and optimization*, which is fine in low dimensional ($x$) continuous problems.
> > >
> > > However, in high dimensional ($x$) and discrete MOO problems (like protein optimization), we argue that scalarization is less practical because it implies learning a new generative model/reparameterization for *each* new scalarization --- which is potentially costly, complex and could be inherently less precise than the continuous setting. This is where an amortized generative model, $q(x|u)$, makes sense. We trade-off some potential precision in preferences for avoiding re-learning a whole new generative model/reparameterization. Additionally, we could use our alignment model, $p(a|x, u=u\_\star)$, to help prune samples from $q(x|u=u\_\star)$ that do not conform to our preferences, $u\_\star$.
> > >
> > > Now to your point on higher dimensional $y$ scaling and our method --- we would say that the real limitation in settings where $L \geq 5$ is actually practically eliciting preferences from a decision maker where cognitive biases begin to dominate, e.g. see [Ruiz et al. 2015], which would affect all methods under consideration.
> > >
> > > These are important considerations in the design of A-GPS, where to use it, and also the limitations of A-GPS and other methods. We will add a discussion point to Section 3.1 and add a full discussion to the appendix of these considerations. We didn't realise this was not coming through in our paper, thank you for pointing this out.
> > >
> > > ---
> > >
> > > [Ruiz et al. 2015] Ana B. Ruiz, Karthik Sindhya, Kaisa Miettinen, Francisco Ruiz, Mariano Luque. E-NAUTILUS: A decision support system for complex multiobjective optimization problems based on the NAUTILUS method, European Journal of Operational Research, Volume 246, Issue 1, 2015, Pages 218-231.

---

> > > > ### Author Response · Authors · 2025-08-06
> > > >
> > > > Thank you again, VcZN, for your insightful feedback.
> > > >
> > > > We would like to double check if you still have any remaining concerns we can clarify while we are still in the discussion phase?
> > > >
> > > > Otherwise, our publication plan for incorporating your feedback is as follows:
> > > >
> > > > 1. Document the adjustments to the algorithm concerning training $q(u)$ and the annealed Pareto rank threshold labelling scheme. All of our (updated) experimental results already incorporate this change. We will also add ablations of these to the appendix.
> > > > 2. Include timings for all methods in the appendix along with existing timings for A-GPS (done).
> > > > 3. Include the two new higher-dimensional synthetic function experiments ($L \geq 5$), and the visualisations of the PCA-projected controlled generation for them (done). Include discussion/limitations of scalarizations vs. amortisation in sections 3.1, 7 and the appendix (including scaling concerns and asymptotics).
> > > >
> > > > Please let us know if we have missed anything! Thank you sincerely for helping us to improve this work.

---

> > > > > ### Author Response · Authors · 2025-08-07
> > > > > **Ablation results**
> > > > >
> > > > > Just following up with your request, we have included some requested ablation results below on the Ehrlich vs. Naturalness experiment ($M=32$) testing $q(u)$ vs $q(u|z)$ for the initial (training) round, and different starting thresholds for the annealing Pareto rank labelling method. These are the results for hyper volume improvement at $t=10$. $p_T = 0.99$ in all cases (i.e. the labels correspond are the Pareto set).
> > > > >
> > > > > | Method | $t=0$ distribution | $p\_0 = 0.01$ | $p\_0 = 0.25$ | $p\_0 = 0.5$ | $p\_0 = 0.75$ | $p\_0 = 0.99$ |
> > > > > |- |- |- |- |- |- |- |
> > > > > | A-GPS LSTM | $q(u\|z)$ | 1.550 (0.185) | 1.630 (0.147) | 1.607 (0.332) | 1.502 (0.165) |  1.385 (0.220) |
> > > > > | | $q(u)$ | 1.472 (0.240) | 1.523 (0.280) | 1.514 (0.155) | 1.578 (0.188) | 1.352 (0.202) |
> > > > > | A-GPS TFM | $q(u\|z)$ | 1.432 (0.200) | 1.552 (0.134) | 1.537 (0.130) | 1.479 (0.206) | 1.509 (0.276) |
> > > > > | | $q(u)$ | 1.430 (0.120) | 1.468 (0.211) | 1.442 (0.119) | 1.524 (0.290) | 1.379 (0.199) |
> > > > >
> > > > > These are interesting! We can see that at the extremes $p\_0 = 0.01$ (mostly positive labels) and $p\_0 = 0.99$ (just the Pareto set as in the original submission) we can see performance dropping off, otherwise there does not to be much difference. Also, suprisingly, generally sticking to $q(u\|z)$ for all rounds seems to offer slightly better performance. These results reinforce your original point, but also shows the new Pareto ranking strategy addresses it (with better initial exploration).
> > > > >
> > > > > All of our current experimental results implement Pareto rank labelling with $p\_0 = 0.75$. Thank you for this suggestion, the main experiments already have been updated (and documented in our rebuttals), and we will add this ablation to the appendix.
> > > > >
> > > > > Thank you again for this suggestion.

---

### Official Review · Reviewer_wcJY · 2025-07-01

**Clarity:** 2
**Significance:** 2
**Originality:** 3
**Rating:** 4
**Confidence:** 3

**Summary:**

The paper proposes a new framework for online multi-objective optimization called active generation of Pareto sets. The method builds upon ideas introduced in variational search distributions, with the goal of extending to multi-objective optimization. The method learns a generative model and uses a class probability estimator, where class is Pareto optimality, to condition this model along with user preferences. This model also does not require expensive hypervolume calculation, which is common in other multiobjective optimization techniques. The authors detail the methodology used and show the results of their method compared to other multi-objective optimization algorithms on a few test functions. These results show strong performance over the test functions presented and flexibility with regard to user preferences and type of generative model.

**Questions:**

Each evaluation appears to have detailed setups for both the baselines and proposed method, which differ across test functions. For example, the model sometimes uses an LSTM and sometimes uses a transformer. However, the performance varies widely across test functions. What is the justification for these decisions and do they provide fair comparison against the baselines? How would you use A-GPS to approach a new problem apriori, including deciding which model to use?

Can you provide further test results? The results shown are fairly limited and a wider range of evaluations would provide more convincing evidence of A-GPS performance. It is mentioned that some test functions have large variations between runs and we do see high standard deviations in the results. Showing further results from other DTLZ and ZDT test problems would be more convincing and help show the true performance of A-GPS.

**Ethical Concerns:**

["NO or VERY MINOR ethics concerns only"]

**Final Justification:**

Updated my score from 3->4 due to the additional results and clarifications from the authors during the rebuttal.

**Limitations:**

The authors mention that specifying hyper-parameters can have a significant impact on performance, but could further expand on how these results change and other possible limitations the methodology has. It appears the selection of these hyperparameters has a significant impact on performance and may be highly tuned for the presented results.

**Quality:**

2

**Strengths And Weaknesses:**

The paper includes good insight into the ability to control user preferences and flexibility within a generative model-based optimization scheme. The results shown in Figure 2 illustrate the ability to condition on user preferences while also matching or beating the hyper-volume performance of baselines. This is particularly helpful given a trained model can be amortized, as users can input different preferences afterwards given a single trained model. This provides an advantage over other methods such as VSD, which do not support this type of amortization. Additionally, the experiments show comparable results from using a generative model as opposed to a GP-based method, which is a promising aspect of the paper.

The experiments shown are relatively limited and would be more convincing with more diversity and quantity of test functions. For example, the DTLZ and ZDT test suites contain further problems with varying pareto frontiers. Additionally, the paper would be improved with clearer discussion of its limitations. The authors mention the limitation of hyperparameter tuning, but it is not clear to what extent this is done for the included results. The results shown require selecting the best model, either transformer or LSTM, for individual test problems, suggesting the robustness of the method could be improved, as the results vary across test functions with the same setup. Therefore, if a single generative model is used with A-GPS, the results may be worse than baselines. As a whole, the paper would be improved by showing how these parameters and test setups were chosen to provide fair comparison to other methods.

---

> ### Author Rebuttal · Authors · 2025-07-31
>
> We sincerely thank reviewer wcJY for the constructive feedback, and we will endeavour to respond to all of their concerns.
>
> **Q1: The experiments shown are relatively limited and would be more convincing with more diversity and quantity of test functions. For example, the DTLZ and ZDT test suites contain further problems with varying pareto frontiers**
>
> Reviewer VCzN also requested additional synthetic function experiment, but with higher dimensional objectives. As such we have tested two more, higher dimensional, synthetic functions, along with another low dimensional GMM test function from BoTorch:
>
> | Test function ($L$) | Method | Mean round 10 relative HVI (1 std.) |
> |-|-|-|
> |DTLZ7 ($L=6$)| A-GPS | 3.073 (1.363) |
> | | qParEGO | 2.313 (1.205) |
> |DTLZ2 ($L=5$)| A-GPS | 1.053 (0.022) |
> | | qParEGO | 1.042 (0.027) |
> |GMM ($L=2$)| A-GPS | 1.057 (0.039) |
> | | qParEGO | 1.055 (0.053) |
> | | qEHVI | 1.105 (0.041) |
> | | qNEHVI | 1.157 (0.024) |
>
> Unfortunately the EHVI-based methods were not able to be used in the higher-dimensional settings (qNEVHI is claimed to scale to $L=5$, but we found difficulty running it in time for this rebuttal), which hopefully reinforces the efficacy of our method.
>
> **Q2: Additionally, the paper would be improved with clearer discussion of its limitations.**
>
> We apologise for not elaborating on the methods limitations. We did slightly expand on the limitations in the supplementary material (Appendix A), but we will further expand the discussion to include:
>
> - A-GPS does not perform well on the GMM test function, we will include this result and attempt to explain the reason (still under investigation).
> - Miss-specification of prior, including overfitting, or poor generative model choice. This is something we have noticed in high dimensional settings, especially with small initial training datasets.
> - Extreme class imbalance when only a few Pareto-optimal solutions exist can lead to overly exploitative behaviour. We have subsequently addressed this --- see the discussion with reviewer VCzN.
> - Incoporating constraints on the design space is less straight-forward with generative models compared to direct design optimization.
> - A possible future work direction is to enhance the precision of the conditional generation as noted by reviewer yJhX.
>
> **Q3: Each evaluation appears to have detailed setups for both the baselines and proposed method, which differ across test functions. For example, the model sometimes uses an LSTM and sometimes uses a transformer. However, the performance varies widely across test functions.**
>
> We should clarify --- in both the Ehrlich vs. Naturalness and RaSP vs. Naturaless experiments we always use both Transformer and LSTM generative models (we do not select between them depending on the experiment).
>
> We do notice in the original experiments that the performance can differ greatly between these generative models. After some digging, we realise that our prior fitting method was too naive, and could cause the Transformers to overfit, especially with such little training data. Subsequently, we have changed the procedure to include 20% dropout (deactivated for the A-ELBO phase), and use 10% held out data to assess convergence. This has lead to much more consistent behaviour, where the transformers and LSTM models perform similarly for shorter sequences ($M \leq 32$), and Transformers generally pull ahead of LSTMs for longer sequences ($M>64$). You can see evidence of this improved consistency in the sample ablation results requested by reviewer uzmJ. Our recommendation to practitioners would be to use Transformers for sequence design tasks so long as they are well validated.
>
> **Q4: How would you use A-GPS to approach a new problem apriori, including deciding which model to use?**
>
> This is something we will be happy to include. We have used the same configuration for all synthetic test functions, and it seems to generalise decently. For high dimensional sequences, we do have a few recommendations, e.g.:
>
> - Generally transformers are better generative models (and more numerically stable) than LSTMs, when properly initialised.
> - We find the sequence length specific configurations given by the VSD authors work well for these problems too.
> - Our CNN CPE architecture seems to perform well with minimal reconfiguration for all sequence problems we have attempted. For longer sequences we suggest larger kernels for the CNN. Ensembles generally to not appear to make much of a performance difference.
> - Make sure the individual components are well validated on the training data where possible --- if anything, slight over-fitting the CPEs leads to preferable outcomes compared to under-fitting, as the prior provides additional regularisation.
> - Using round-varying Pareto-rank based labelling (see discussion with reviewer VCzN) makes the method more robust to cases where there are only a few Pareto-optimal solutions. We typically start with a threshold of the 75th-percentile of ranks, and by t=T, end with the Pareto set.
>
> We will add this to the paper, or at least to the appendix and reference it in the main text.

---

> > ### Comment · Reviewer_wcJY · 2025-08-08
> > **Thank you for the response**
> >
> > Dear Authors, I'd like to thank you for your detailed rebuttal and for providing additional experimental results, clarifications, and discussion of limitations.
> >
> > I appreciate the inclusion of further test functions (DTLZ7, DTLZ2 with higher dimensions, and GMM) and agreeing to expand the limitations section, as these are super important additional materials to be provided in the revised paper. Note that qParEGO is generally considered a weak baseline, so it's worth spending some extra effort to get qEHVI work (though engineering, simplification, or hyper-param tuning) or compare with some more advanced baselines.
> >
> > The clarification regarding model choice (Transformer vs. LSTM) and the adjustments to reduce overfitting are also helpful, as is the added guidance on how practitioners might approach applying A-GPS to new problems. It's unfortunate that they were not properly diagnosed before the submission, but I believe they will be very helpful in revising the paper.
> >
> > At this point, I don't need extra clarification, and thanks again for the detailed information.

---

> ### Author Response · Authors · 2025-08-05
>
> Esteemed reviewer wcJY,
>
> We referred to additional results to other reviewers in our response to you. To aid your assessment of our work, we have now copied them here for your convenience.
>
> **Q3: Each evaluation appears to have detailed setups for both the baselines and proposed method, which differ across test functions. For example, the model sometimes uses an LSTM and sometimes uses a transformer. However, the performance varies widely across test functions**
>
> > You can see evidence of this improved consistency in the sample ablation results requested by reviewer uzmJ
>
> We replicate these below (round t=10 relative hyper-volume improvement mean and std. dev.):
>
> | Seq. Len | Method | $S=64$ | $S=128 $ | $S=256$ | $S=512$ |
> | -| -| -| -| -| -|
> | 32 | AGPS-TFM | 1.600 (0.307) | 1.576 (0.119) | 1.567 (0.241) | 1.584 (0.125)
> | | AGPS-LSTM | 1.563 (0.323) | 1.484 (0.225) | 1.597 (0.290)| 1.514 (0.267)
> 64 | AGPS-TFM | 1.071 (0.058) | 1.094 (0.047) | 1.070 (0.058) | 1.094 (0.050)
> | | AGPS-LSTM | 1.088 (0.048) | 1.075 (0.053) | 1.075 (0.052) | 1.082 (0.048)
>
> **Q1: The experiments shown are relatively limited and would be more convincing with more diversity and quantity of test functions.**
>
> In addition to the extended synthetic test functions we presented, we also presented the following extended sequence optimisation results to reviewer uzmJ.
>
> > Futhermore, we have also implemented the Solvabity vs. SASA experiment from [31] which uses the Foldx simulator as a black box. This experiment is run for 64 rounds, with batches of 16, and only permits one mutation per sequence per round. This required a masked Transformer generative model (mTFM), $q$, that learns to mutate an input sequence, in a similar fashion to the generative back-bones of LaMBO and LaMBO-2. A-GPS and VSD perform similarly to LaMBO-2 on this experiment (and outperform a greedy random baseline and CbAS), and we feel it adds yet another compelling example of this method. Round 64 results summarised below,
>
> | Method | Mean round 10 relative HVI (1 std.) |
> | -| -|
> | VSD-mTFM | 1.150 (0.021)
> | LaMBO2 | 1.142 (0.025)
> | A-GPS-mTFM | 1.137 (0.025)
> | CbAS-mTFM | 1.078 (0.025)
> | Random (greedy) | 1.061 (0.013)
>
> Thank you for your appraisal of our work.

---

> ### Author Response · Authors · 2025-08-08
>
> Esteemed reviewer -- we have managed to produce results for qNEHVI for $L=5$ below:
>
> | Test function ($L$) | Method | Mean round 10 relative HVI (1 std.) |
> |-|-|-|
> |DTLZ2 ($L=5$)| A-GPS | 1.053 (0.022) |
> | | qParEGO | 1.042 (0.027) |
> | | qNEHVI | 1.036 (0.019) |
>
> We did have to decrease the number of random restarts from 10 to 5, and reduce the number of samples to approximate EHVI from 128 to 64, and so this should viewed as a lower bound result. Furthermore, reviewer uzmJ has requested we change the experiment for the GP-based methods are optimising in the original, $[0, 1]^d$, space, and not the sigmoid transformed space. We are currently working on reproducing these synthetic test function results to honour this request.

---

### Official Review · Reviewer_uzmJ · 2025-07-02

**Clarity:** 4
**Significance:** 3
**Originality:** 3
**Rating:** 5
**Confidence:** 4

**Summary:**

This paper proposes a novel approach to multi-objective optimization (MOO) by directly modeling the Pareto front, rather than relying on traditional model-based methods. This strategy avoids common pitfalls associated with hypervolume-based approaches, offering a more stable and efficient alternative that also works seamlessly for discrete problems. A key strength of the method is its ability to incorporate user preferences into the exploration of the Pareto front without requiring retraining of the Pareto classifier. Empirical results demonstrate strong performance across a representative set of baselines, highlighting the method’s practical effectiveness.

**Questions:**

Please can you provide a more detailed explanation for Fig 1(a) ?
Please can you provide detailed info about the baseline methods you ran in BOTORCH?
Please can you comment on why you used a sigmoid transform for the domains of the synthetic test problems? Did you then use Botorch in the unbounded space? This feels like it will impair the standard MOBO approaches.

**Ethical Concerns:**

["NO or VERY MINOR ethics concerns only"]

**Final Justification:**

The comprehensive rebuttal dealt with my concerns (noteably around repoducibility and ablation) and I raised my score. I think this work poses an intersting avenue for MOO and is well backedup with extensive emperical results.

**Quality:**

3

**Strengths And Weaknesses:**

I really enjoyed reading this paper. As someone with expertise in Bayesian optimization but less familiarity with multi-objective generation (MOG) approaches, I found the paper informative and very accessible. The concept of active generation is clearly defined, and the overall presentation is strong. However, I found Figure 1(a) a bit hard to interpret and would have appreciated a more detailed caption—this figure plays a central role in conveying the method’s intuition. In contrast, Figure 1(b) added relatively little. While I don’t feel qualified to assess the significance of the contribution relative to prior MOG methods, I do think the work is well-executed and worthwhile. Thank you for a thoughtful and well-written paper.

My main concerns lie in the experimental section. Since the method lacks strong theoretical justification, a rigorous empirical evaluation is critical for a venue like NeurIPS. First, I would have liked to see an ablation that justifies the use of the preference alignment component described in lines 148–153—this was the one part of the method I found difficult to follow, and further evidence of its impact would be valuable. Additional ablations (e.g., on the number of samples used to approximate Equation 18) would also strengthen the paper. More seriously, a lot of implementation details for the baselines (mentioned on line 230) are missing entirely (e.g. how are acquisition functions optimized, choices of reference points e.t.c). In Bayesian optimization and MOO, these details are crucial, as implementation nuances can heavily affect performance. Without this information, it’s difficult to assess the validity of the performance comparisons. I also found it concerning that Figure 2 is based on only five repetitions—this raises questions about the statistical robustness of the results. That said, Sections 6.2 and 6.3 were compelling, and I appreciated the inclusion of more challenging, real-world examples to demonstrate the method’s applicability.

---

> ### Author Rebuttal · Authors · 2025-07-31
>
> We thank reviewer uzmJ for the thoughtful and considered review, and we are encouraged they enjoyed reading the paper! We will attempt to address the reviewer's concerns succinctly.
>
> **Q1: Please can you provide a more detailed explanation for Fig 1(a).**
>
> Thank you for this feedback. We will consider moving Fig 1(b) to the appendix, or shortening its description since it is a visual representation of Alg. 1.
>
> We will change the description of Fig 1(a) to something like the following: (a) A visualisation of a 2-dimensional Pareto front, $\mathcal{F}\_\text{Pareto}$, and the random variables used with A-GPS. $y$ are the noisy realisations of the objectives, $f$, when $z=1$ these observations lie on the Pareto front. Preference direction vectors, $u$, are simply unit vectors pointing to a region of the Pareto front from a reference point, $r$ --- Equation 9. We derive aligned ($a=1$) training preference direction vectors, $u_n$ from observation pairs $(y_n, x_n)$, and mis-aligned preference direction vectors ($a=0$) from permuting these pairs, $(y_{\rho(n)}, x_n)$ --- Equation 16. The aim is to learn the distribution of the Pareto set $q_\phi(x|u) \approx p(x|u, z=1, a=1)$, from which a user can input a preference encoded as $u$, for generating optimal samples. We show the image of this distribution on the Pareto front.
>
> **Q2: Please can you provide detailed info about the baseline methods you ran in BOTORCH? Please can you comment on why you used a sigmoid transform for the domains of the synthetic test problems? Did you then use Botorch in the unbounded space? This feels like it will impair the standard MOBO approaches ...**
>
> Many of these details are in the Appendix D included in the supplementary material along with the original submission, including, e.g. GP surrogate model, reference points for the problems, and details on the sigmoid transform (which is treated as part of the black box). We will expand on these for the final version.
>
> The reference points used are supplied by the BoTorch library implementations of the black boxes where possible --- though in one case we found the supplied value erroneous. We have actually found that using the BoTorch `botorch.utils.multi_objective.hypervolume.infer_reference_point` function is a more reliable way of setting the reference point automatically, and it is used by other methods, e.g. LaMBO-2. So all of our experiments, methods and baselines have been adapted to use this method now.
>
> All of the original synthetic test functions are only defined on $[0, 1]^D$, and so composing these black-boxes with a sigmoid function is an easy way of making them unbounded, freeing us to use more expressive generative models. The GP-based methods also optimize within this unbounded space --- though we do give them generous bounds of $[-15, 15]^D$, since the BoTorch interface requires bounds to be specified. We believe this is a fair comparison.
>
> We will happily expand upon these details where the reviewer sees fit. We will add these details for the GP methods:
>
> | Property | Value |
> |--------|------|
> | Optimizer | Adam |
> | Iterations | 500 |
> | # Restarts | 10 |
> | Bounds | $[-15, 15]^D$ |
> | Kernel | Matern $\nu = 2.5$ ARD (standardized inputs) |
> | Hyperparameters | LogNormal priors on $\sigma$ and $l$ |
>
> Are there any other settings in particular we should add?
>
> **Q3: I would have liked to see an ablation that justifies the use of the preference alignment component described in lines 148–153**
>
> We would like to clarify to the reviewer that the preference alignment indicators are *necesary* as one of the contributions of our approach is to support user preferences at test time. In order to train such a model, we need a way to tell whether a sample $x_i$ is indeed aligned with preference vector $u_i$. For this purpose, we use an alignment indicator $a_i$. Both preference direction vectors and alignment indicators are constructed internally by our framework (as described in the paper) during training so that our conditional generative model $q(x | u)$ learns to generate Pareto-optimal samples that are aligned with preferences. At test time, a user can specify a new preference vector $u_\star$ and our model will generate samples $\{x_\star\}$ that are aligned as much as possible with such preferences. We can add this clarification to the text.
>
> **Q4: Additional ablations (e.g., on the number of samples used to approximate Equation 18) would also strengthen the paper.**
>
> We will happily include these. Here are round ten, $t=10$, results for different sample sizes, $S$, on the Ehrlich vs. Naturalness experiment,
>
> | Seq. Len | Method | $S=64$ | $S=128 $ | $S=256$ | $S=512$ |
> | -| -| -| -| -| -|
> | 32 | AGPS-TFM | 1.600 (0.307) | 1.576 (0.119) | 1.567 (0.241) | 1.584 (0.125)
> | | AGPS-LSTM | 1.563 (0.323) | 1.484 (0.225) | 1.597 (0.290)| 1.514 (0.267)
> 64 | AGPS-TFM | 1.071 (0.058) | 1.094 (0.047) | 1.070 (0.058) | 1.094 (0.050)
> | | AGPS-LSTM | 1.088 (0.048) | 1.075 (0.053) | 1.075 (0.052) | 1.082 (0.048)
>
> Interestingly, for the LSTM and Transformer we do not find significant variability -- we will expand this to other experiments.
>
> **Q5: I also found it concerning that Figure 2 is based on only five repetitions --- this raises questions about the statistical robustness of the results.**
>
> We will increase this to 10 replicates for the final version. We have implemented this already, and found little difference in the conclusions drawn, apart from improving qParEGO on DTLZ2 --- here are round 10 results presented as relative hyper-volume improvement:
>
> | Test function ($L$) | Method | Mean round 10 relative HVI (1 std.) |
> |-|-|-|
> |BraninCurrin ($L=2$)| A-GPS | 1.626 (0.085) |
> | | qParEGO | 1.374 (0.133) |
> | | qEHVI | 1.546 (0.092) |
> | | qNEHVI | 1.510 (0.107) |
> |DTLZ2 ($L=2$)| A-GPS | 1.071 (0.020) |
> | | qParEGO | 1.042 (0.027) |
> | | qEHVI | 1.060 (0.004) |
> | | qNEHVI | 1.060 (0.008) |
> |ZDT3 ($L=2$)| A-GPS | 1.267 (0.312) |
> | | qParEGO | 1.041 (0.008) |
> | | qEHVI | 1.045 (0.044) |
> | | qNEHVI | 1.082 (0.107) |
>
>
> **Q6: Since the method lacks strong theoretical justification, a rigorous empirical evaluation is critical for a venue like NeurIPS ... That said, Sections 6.2 and 6.3 were compelling, and I appreciated the inclusion of more challenging, real-world examples to demonstrate the method’s applicability.**
>
> Subsequent to the paper's submission we have realised that under certain assumptions the CPE trained on non-dominance indicators is estimating probability of hyper-volume improvement (PHVI). We have sketched the proof for reviewer yJhX. Hopefully including this will improve the method's theoretical justification.
>
> Futhermore, we have also implemented the Solvabity vs. SASA experiment from [31] which uses the Foldx simulator as a black box. This experiment is run for 64 rounds, with batches of 16, and only permits one mutation per sequence per round. This required a masked Transformer generative model (mTFM), $q$, that learns to mutate an input sequence, in a similar fashion to the generative back-bones of LaMBO and LaMBO-2. A-GPS and VSD perform similarly to LaMBO-2 on this experiment (and outperform a greedy random baseline and CbAS), and we feel it adds yet another compelling example of this method. Round 64 results summarised below,
>
> | Method | Mean round 64 relative HVI (1 std.) |
> | -| -|
> | VSD-mTFM | 1.150 (0.021)
> | LaMBO2 | 1.142 (0.025)
> | A-GPS-mTFM | 1.137 (0.025)
> | CbAS-mTFM | 1.078 (0.025)
> | Random (greedy) | 1.061 (0.013)

---

> > ### Comment · Reviewer_uzmJ · 2025-08-04
> >
> > Thanks for this comprehensive rebuttal that dealt with my concerns. I have raised my score.

---

> > > ### Author Response · Authors · 2025-08-04
> > > **thank you**
> > >
> > > Esteemed Reviewer,
> > > \
> > > \
> > > We thank you for engaging with our rebuttal, and for valuable comments. Rest assured we will incorporate all requested modifications into our paper.
> > > \
> > > \
> > > Meantime, if there is anything else we can improve or answer, kindly do let us know.
> > >
> > > Best regards,
> > >
> > > Authors

---

> > > > ### Comment · Reviewer_uzmJ · 2025-08-07
> > > >
> > > > I want to stress that I really enjoyed reading the addition of the proofs showing that Pareto CPE is a cheap approximation of PHVI from the dicussion with one of the other reviewers. I look forward to reading this in the final version.

---

> > > > > ### Author Response · Authors · 2025-08-07
> > > > > **thank you**
> > > > >
> > > > > Absolutely,
> > > > > \
> > > > > \
> > > > > We are very glad that proofs re. Pareto CPE = cheap approximation of PHVI are useful. We hope other reviewers will also appreciate this result.
> > > > > \
> > > > > \
> > > > > This feedback is much appreciated.
> > > > >
> > > > > Authors.

---

> > > > > ### Comment · Reviewer_uzmJ · 2025-08-08
> > > > > **Additional concern**
> > > > >
> > > > > The proposed figure caption is much clearer (although somewhat verbose).  I did see Appendix D, but this is still lacking many of the important details. Its critical to know that you have put in similar effort in acquisition optimisation across the baselines (as it is very easy to hamstring these methods through inappropriate choices). Thanks for the info about reference points — this is also critical to include in the paper. I still very much worry about the fact that you fit GPs in the sigmoid space, as this will ruin smoothness properties that the GP relies on and isnt at all a realistic implementation of BO. If your problem is defined on a box [0,1]^D, then you need to run your BO and surrogate modelling on that space! I worry that this is an unfair comparison.

---

> > > > > > ### Author Response · Authors · 2025-08-08
> > > > > >
> > > > > > Thank you for your continued feedback,
> > > > > >
> > > > > > We will make the caption more concise for figure 1a, but also move 1b to the appendix to allow for more description of 1a as we feel it is important for helping readers develop an intuition for the problem.
> > > > > >
> > > > > > Re Appendix D -- yes, we will also add all of the details we presented in response to Q2 above, and any others you would like. We have also provided our code (and intend to release it as OSS). We want to ensure A-GPS is practially useful, and so we are concerned in portraying it honestly in comparison to well established, and implemented, baselines.
> > > > > >
> > > > > > A sigmoid function is infinitely differentiable, and so smooth. But perhaps you are concerned about the behaviour of shift-invariant kernels in a warped space, e.g. near the bounds large shifts in the warped space only imply increase small shifts in the original space? This could result in the Matern kernels of the GPs performing worse near these bounds, but the NNs used by A-GPS will also experience the same difficulties (though they are not stationary).
> > > > > >
> > > > > > At the moment we are treating the sigmoid function as _part of the black-box_. We can make this more obvious in the paper. E.g. label the figures "sigmoid transformed BraninCurrin" or "Sigmoid $\circ$ BraninCurrin" and make it explicit in the text. Would this help your concern, so a direct cross comparison to other works has this context?
> > > > > >
> > > > > > Or, we can optimise the GPs in the original space, but A-GPS in the sigmoid space (with a sigmoid transformed reference point)? This is also potentially problematic as now different "black-boxes" are being optimised, but it makes for easier comparison across papers.
> > > > > >
> > > > > > We can see the advantage of both ways forward. We would appreciate any advice you have on how to best resolve this situation. Thank you.

---

> > > > > > > ### Comment · Reviewer_uzmJ · 2025-08-08
> > > > > > >
> > > > > > > I would suggest runing the BO/GPs in the original space and your aproach in the sigmoid space. This is the fair comparision I think. Its a "weakness" (or "quirk") of your setup that means that you have to use the sigmoid, and so you shouldnt impair the BO baslines with.

---

> > > > > > > > ### Author Response · Authors · 2025-08-08
> > > > > > > >
> > > > > > > > Thank you for your input.
> > > > > > > >
> > > > > > > > We will go with your suggestion, and run the GP-based methods in the original $[0, 1]^d$, and only use the sigmoid transform for A-GPS's generative model.
> > > > > > > >
> > > > > > > > We will endeavour to re-run all experiments, but at this late stage we may not get them all ready in time.
> > > > > > > >
> > > > > > > > We would also like to reinforce that A-GPS was actually originally designed for high-dimensional and discrete optimisation problems, and to us it is a bonus that it is even competitive with EHVI methods on continuous MOBO problems (where having a generative model adds complexity). But, we will make sure _all_ these experiments are fair and thorough none-the-less.

---

> ### Author Response · Authors · 2025-08-09
>
> We have managed to re-run some of the synthetic function experiments with the GPs in $[0, 1]^d$. Unfortunately, this meant re-defining the initial data, $\mathbf{x}_0$, as being sampled from Latin hypercube sampling in $[0, 1]^d$, and not in the sigmoid space, as it was originally (with bounds of $[-1, 1]^d$). In our haste, we forgot to re-fit AGP's prior to this new data, and so it has been run with suboptimal settings (prior being too concentrated). You can see this when comparing these results to those reported previously (though BraninCurrin uses a different scale, and so is not comparable). We don't have time to re-run these experiments, so we will just report what we have as a gesture of good faith:
>
> | Test function ($L$) | Method | Mean round 10 relative HVI (1 std.) |
> |-|-|-|
> |BraninCurrin ($L=2$)| A-GPS | 1.160 (0.023) |
> | | qParEGO | 1.055 (0.022)|
> | | qEHVI | 1.170 (0.005) |
> | | qNEHVI | 1.176 (0.004)) |
> |DTLZ2 ($L=2$)| A-GPS | 1.017 (0.004) |
> | | qParEGO | 1.029 (0.004) |
> | | qEHVI |  1.052 (0.001) |
> | | qNEHVI | 1.053 (0.001) |
> |ZDT3 ($L=2$)| A-GPS | 1.133 (0.184) |
> | | qParEGO | 1.126 (0.177) |
> | | qEHVI | 1.153 (0.218) |
> | | qNEHVI | 1.155 (0.173) |
> |DTLZ7 ($L=6$)| A-GPS | 1.480 (0.413)|
> | | qParEGO |1.792 (0.454) |
> |GMM ($L=2$)| A-GPS | 1.030 (0.025) |
> | | qParEGO | 1.181 (0.020) |
> | | qEHVI | 1.217 (0.002) |
> | | qNEHVI | 1.217 (0.002) |
>
> Unfortunately DTLZ2 with $L=5$ did not finish in time.

---

### Comment · Area_Chair_iqCN · 2025-08-04

Hi reviewers, if you have not yet engaged with the authors' response (thank you Reviewer yJhX for taking the lead), please raise any remaining questions and/or concerns so the authors have a chance to reply.

---

### Note · Authors · 2025-08-11

We thank all reviewers for their constructive feedback, and are encouraged by the consensus that A-GPS is a novel contribution. In discussions, we further increased its theoretical strength by proving that the Pareto CPE estimates the *PHVI* under general conditions.

While much attention focused on continuous synthetic test functions, A-GPS is primarily designed for **high-dimensional, discrete MOO** - where direct optimisation is infeasible, few effective methods exist, and real-world applications (e.g. protein design) are clear. The synthetic functions serve mainly as proof-of-concept; nonetheless, we expanded these experiments, ensured fairer baselines, and added higher-dimensional benchmarks.

**Following up with uzmJ**: with the GPs operating in $[0, 1]^d$ and further refined, A-GPS now ranks similarly to qParEgo in performance.

**To address potential concerns from yJhX**: we claim the revisions are **minor** -- extra ablations (done), refined experiments (done), and textual clarifications (planned),

| Reviewer | Key Requests | Proposed Revisions | Revision Effort |
|----------|-------------|--------------|-----|
| uzmJ | More theory; ablations; GP baseline details & improvements | Add PHVI proof; insert new ablations (completed -- alignment, A-ELBO samples); insert BO details; increased replicates & no sigmoid transform for GPs (completed) | 1-2 paragraphs for theorem in S2.1, abstract change; appendix: extra experiments & detail, full proof
| wcJY | More diverse test functions; model choice clarity; limitations | Add DTLZ7, high-dim DTLZ2, GMM (completed); clarify Transformer vs. LSTM usage; expand limitations | Put new results in S6.1 & appendix; ~5-6 more lines to Discussion; appendix material on A-GPS use (guide) and limitations
| VCzN | Better Pareto coverage; runtime breakdown; higher-dim results | Add rank-threshold labelling description; provide runtime tables; include high-dim DTLZ results (completed) | ~4 more lines in main text (around S2.1 & eqn. 4, after proof); Modify Alg. 1; appendix material
| yJhX | Alignment training justification; baseline correctness; clarify preference usage; clarify amortisation | Add alignment justification and negative-label upsampling; fixed LaMBO-2 bug & CbAS improved (completed); clarify preference use & amortisation | ~5 more lines in S4.1; minor rewording re amortisation; appendix guide

We thank the reviewers for their engagement that has resulted in a stronger, clearer, and more rigorous paper.

---

### Decision · Program_Chairs · 2025-09-17

**Decision:**

Accept (poster)

**Comment:**

This paper proposes Active Generation of Pareto Sets (A-GPS), a framework for online discrete black-box multi-objective optimization (MOO) that learns a generative model of the Pareto set supporting a-posteriori preference conditioning. The key scientific claims include: (1) A-GPS can effectively amortize over user preferences without requiring retraining; (2) The method avoids expensive hypervolume computation through a class probability estimator (CPE) for Pareto-optimality prediction; (3) A-GPS demonstrates competitive performance against established baselines like qEHVI and qNEHVI on both synthetic and real-world tasks including protein design benchmarks.

Strengths:
- Motivation and relevance
- Originality + novel methodological contributions
- Incorporation of preferences via conditional generative model is interesting and useful
- Technical soundness
- Empirical performance (with caveats due to some issues with evaluation)

Weaknesses:
- Empirical evaluation limited in terms of problems, baselines, and ablation studies (partially addressed during rebuttal)
- Insufficient discussion of implementation details

Reviewers generally agreed on the strong motivation and relevance of the work and the clarity of the presentation. They highlighted the novelty of the approach and the corresponding methodological contributions, arguing that while the general ideas of the paper rely on previous work by Steinberg et al. (2025), the specific setting in the paper considers additional challenges and extensions that go substantially beyond that work. They also found the performance of the method to be promising (subject to the caveats below).

Reviewers’ concerns concentrated on the lack of rigor in the empirical evaluation, highlighting the limited coverage of problems and baselines, missing ablation studies, lack of statistical significance of some results, insufficient discussion of implementation details, and some concerns about implementation quality. In turn, the authors provided a comprehensive rebuttal addressing the different points raised. Reviewers appreciated the detailed response, and found that the authors addressed most of their concerns*, including those about reproducibility and ablations. Reviewers also appreciated the additional proof (sketch) for the equivalence between Pareto CPE and PHVI estimation.

*One thing that I believe has not been sufficiently addressed is the high variance in the experimental results. Even with the increased number of replicates, the differences between the methods are often not statistically significant. I don’t think this is a critical flaw since the paper provides a new approach to the problem (rather than just claiming to improve performance), but it is something that I would like to understand better through additional replications or problems where differences are more clearly pronounced.

Overall, the strength of the paper significantly improved during the rebuttal and author discussion period, and reviewers generally consider the work a meaningful contribution. At this point, the primary question is whether the required changes to the manuscript are substantial enough to require a completely new review cycle. The opinions of the reviewers on this diverge somewhat and I can see both sides of the argument. However, as many if not all of the results on the requested improvements have already been provided as part of the rebuttal there is little risk in trusting the authors to properly incorporate these changes into their final manuscript. Therefore, I believe the submission should be accepted.

That said, I do want to emphasize reviewer yJhX’s excellent point that a proper revision and additional review could “make the difference between an okay paper and a very good paper” and suggest the authors take that to heart when updating their manuscript.